# Classical and Bayesian estimation for type-I extended-*F* family with an actuarial application

Nada M. Alfaer[1]☯, Sarah A. Bandar[2]☯, Omid Kharazmi[3]☯, Hazem Al-Mofleh[4]☯, Zubair Ahmad[5]☯, Ahmed Z. Afify[6]☯ *

1 Department of Mathematics & Statistics, College of Science, Taif University, Taif, Saudi Arabia, 2 Department of Mathematics, College of Education, Misan University, Amarah, Iraq, 3 Department of Statistics, Faculty of Sciences, Vali-e-Asr University of Rafsanjan, Rafsanjan, Pakistan, 4 Department of Mathematics, Tafila Technical University, Tafila, Jordan, 5 Department of Statistics, Quaid-i-Azam University, Islamabad, Pakistan, 6 Department of Statistics, Mathematics and Insurance, Benha University, Benha, Egypt

☯ These authors contributed equally to this work.
* ahmed.afify@fcom.bu.edu.eg

**Data Availability Statement:** All relevant data can be found in the Github repository (https://github.com/almof1hm/Hospital-costs-in-the-state-of-Wisconsin/blob/main/Data.xlsx).

## Abstract

In this work, a new flexible class, called the type-I extended-*F* family, is proposed. A special sub-model of the proposed class, called type-I extended-Weibull (TIEx-W) distribution, is explored in detail. Basic properties of the TIEx-W distribution are provided. The parameters of the TIEx-W distribution are obtained by eight classical methods of estimation. The performance of these estimators is explored using Monte Carlo simulation results for small and large samples. Besides, the Bayesian estimation of the model parameters under different loss functions for the real data set is also provided. The importance and flexibility of the TIEx-W model are illustrated by analyzing an insurance data. The real-life insurance data illustrates that the TIEx-W distribution provides better fit as compared to competing models such as Lindley–Weibull, exponentiated Weibull, Kumaraswamy–Weibull, *α* logarithmic transformed Weibull, and beta Weibull distributions, among others.

## 1 Introduction

Modeling insurance losses data using heavy-tailed distributions is an important subject matter for risk managers and actuaries. Generally, the insurance sets of data are usually right-skewed, hump-shaped, unimodal, and have thick right tail. Distributions possessing such characteristics are considered prominent candidates for modeling such heavy-tailed data.

The heavy-tailed distributions are adopted to estimate insurance losses data and thereby helps in assessing of business risk level. Hence, due to its immense significance in actuarial sciences, these types of data are studied and explored extensively and several distributions are introduced in the actuarial literature.

**Funding:** This study was funded by the Taif University Researchers Supporting 279 Project (TURSP-2020/316), Taif University, Taif, Saudi Arabia. The funders had no role in study design, data collection and analysis, decision to publish, or preparation of the manuscript.

**Competing interests:** The authors have declared that no competing interests exist.

**Abbreviations:** ABs, Absolute bias; ADEs, Anderson-Darling estimates; ALW, $\alpha$ logarithmic transformed Weibull; APT, Alpha-power transformation; AVEs, Average estimates of the parameters; BEs, Bayes estimates; Bur, Burr; BW, Beta Weibull; cdf, Cumulative distribution function; CVMEs, Cramér-von Mises estimates; EW, Exponentiated Weibull; Ex-APT, Extended alpha-power transformation; hrf, Hazard rate function; JPD, Joint posterior density; KMS, Kaplan-Meier survival; K-S, Kolmogorov-Smirnov; Kur, Kurtosis; KwW, Kumaraswamy Weibull; LFs, Loss functions; LiW, Lindley Weibull; Lo, Lomax; LSEs, Least squares estimates; MLEs, Maximum likelihood estimates; MPSEs, Maximum product of spacings estimates; MREs, Mean relative errors; MSELF, Modified SELF; MSEs, Mean-squared errors; PCEs, Percentiles estimates; pdf, Probability density function; PLF, Precautionary LF; qf, Quantile function; RADEs, Right-tail Anderson-Darling estimates; rv, Random variable; SELF, Squared error LF; SEs, Standard errors; Sk, Skewness; TIEx-F, Type-I extended-F; TIEx-W, Type-I extended-Weibull; Wi, Weibull; WLSEs, Weighted least squares estimates; WSELF, Weighted SELF.

The insurance loss data, financial returns, file sizes on network servers etc, are explored and modeled by several models such as the Lomax [1], Pareto [2], Weibull [3], Burr [4], log-logistic [5], Stoppa [6] distributions, among others.

However, the classical distributions are still not flexible enough to adequately fit such data sets, and some distributions do not possess closed form for its cumulative distribution function (cdf) causing difficulties related to the parameter estimation. For example, (i) the Pareto distribution has a monotonically decreasing shape for its probability density function (pdf), and it does not often provide best fit to many data sets, (ii) Weibull distribution is suitable for modeling small losses, but, unfortunately, it is not suitable for modeling large losses, and (iii) log-normal distribution does not have closed form solution of cdf result in estimation consequences.

To overcome the aforementioned problems, the researchers have been working on to propose new distributions to address these issues. For example, the heavy-tailed distributions by [7], Pareto–Levy by [8], loss models by [9], Weibull–Pareto by [3], generalized log-Moyal by [10], and generalized Pareto by [11].

The TIEx-*F* family has some desirable characteristics. It is novel and very simple approach of adding an additional parameter to generalize the existing distributions, hence it can provide extended versions of baseline distributions with closed form expressions for their cdfs and hrfs. The TIEx-*F* family provides better fit than other competing modified models under the same baseline model. Further a new lifetime model, based on the TIEx-*F* family, called the TIEx-Weibull (TIEx-W) distribution is studied. The TIEx-W distribution can provide increasing, decreasing and modified bathtub shaped hrf.

Additionally, the TIEx-W parameters are estimated by using several estimation methods and their performance is explored by detailed simulations for small and large samples. Many authors have addressed different estimators to estimate the parameters of generalized models such as the Weibull–Marshall–Olkin power-Lindley distribution by [12] and the Marshall–Olkin–Weibull exponential distribution by [13].

The rest of this work is unfolded as follows. Section 2 is devoted to introducing the proposed family. In Section 3, we present a special sub-model of the proposed method. Some characteristics are provided in Section 4. Section 5 is devoted to the estimation of the model parameters. A detailed simulation study is explored in Section 6. A real data application is discussed in Section 7. Bayesian estimation under five loss functions are explored in Section 8. The paper is concluded in Section 9.

## 2 The TIEx-F family

In the recent era, the statisticians have shown an increased interest to propose new family of distributions by introducing additional parameters. In this credit, Mahdavi and Kundu [14] pioneered a new method, called the alpha-power transformation (APT) family, for generating univariate distributions using the following cdf

$$G(x; \alpha, \mathbf{\Psi}) = \frac{\alpha^{F(x;\mathbf{\Psi})} - 1}{\alpha - 1}, \qquad \alpha > 0, \mathbf{\Psi} \in \mathbb{R}^k, \alpha \neq 1, x \in \mathbb{R}. \tag{1}$$

The AP-F class is studied with more detail in [15]. [16] proposed another approach, called extended APT (Ex-APT) family for generating new distributions which is specified by the cdf

$$G(x; \alpha, \mathbf{\Psi}) = \frac{\alpha^{F(x;\mathbf{\Psi})} - e^{F(x;\mathbf{\Psi})}}{\alpha - e}, \qquad \alpha > 0, \mathbf{\Psi} \in \mathbb{R}^k, \alpha \neq 1, \alpha > e, x \in \mathbb{R}. \tag{2}$$

[17] introduced another new method called, Ampadu APT family which is defined by the cdf

$$G(x; \alpha, \boldsymbol{\Psi}) = \frac{e^{-F(x;\boldsymbol{\Psi})} - \alpha^{F(x;\boldsymbol{\Psi})}}{e^{-1} - \alpha}, \qquad \boldsymbol{\Psi} \in \mathbb{R}^k, \ \alpha > \frac{1}{e}, \ x \in \mathbb{R}. \tag{3}$$

In this work, we address a new class to generate new flexible distributions, called type-I extended-$F$ (TIEx-$F$) family which has the following cdf

$$G(x; \eta, \boldsymbol{\Psi}) = \frac{e^{F(x;\boldsymbol{\Psi})} - [1 - \eta F(x; \boldsymbol{\Psi})]}{e - \bar{\eta}}, \qquad \eta > 0, \boldsymbol{\Psi} \in \mathbb{R}^k, \eta \neq 1 - e, x \in \mathbb{R}. \tag{4}$$

where $F(x; \boldsymbol{\Psi})$ is cdf of the baseline random variable ($rv$) depending on the vector of parameters $\boldsymbol{\Psi}$, and $\eta$ is an additional parameter.

The pdf corresponding to (4) takes the form

$$g(x; \eta, \boldsymbol{\Psi}) = \frac{f(x; \boldsymbol{\Psi})[e^{F(x;\boldsymbol{\Psi})} + \eta]}{e - \bar{\eta}}, \qquad x \in \mathbb{R}. \tag{5}$$

The new pdf (5) of the TIEx-$F$ family will be most tractable for simple analytical expressions of $F(x; \boldsymbol{\Psi})$ and $f(x; \boldsymbol{\Psi})$. Henceforth, a $rv$ $X$ with pdf (5) is denoted by $X \sim$ TIEx-$F$ ($\eta, \boldsymbol{\Psi}$).

## 3 The TIEx-W distribution

Using the cdf of the Weibull (W) distribution, $F(x; \boldsymbol{\Psi}) = 1 - e^{-\gamma x^\alpha}, \quad , x > 0, \ \gamma, \ \alpha > 0$ and its pdf, $f(x; \boldsymbol{\Psi}) = \alpha \gamma x^{\alpha-1} e^{-\gamma x^\alpha}$, where $\boldsymbol{\Psi} = (\alpha, \gamma)^\top$, we obtain the cdf of the TIEx-W distribution

$$G(x; \alpha, \gamma, \eta) = \frac{e^{(1-e^{-\gamma x^\alpha})} - [1 - \eta(1 - e^{-\gamma x^\alpha})]}{e - \bar{\eta}}, \qquad \alpha, \eta, \gamma > 0, x > 0. \tag{6}$$

The corresponding pdf of the TIEx-W distribution reduces to

$$g(x; \alpha, \gamma, \eta) = \frac{\alpha \gamma x^{\alpha-1} e^{-\gamma x^\alpha}(e^{(1-e^{-\gamma x^\alpha})} + \eta)}{e - \bar{\eta}}, \qquad x > 0. \tag{7}$$

Fig 1 sketches some density plots for the TIEx-W distribution. It can be seen that the pdf shapes of this model can be: reversed-J, left-skewed, right-skewed, or symmetry.

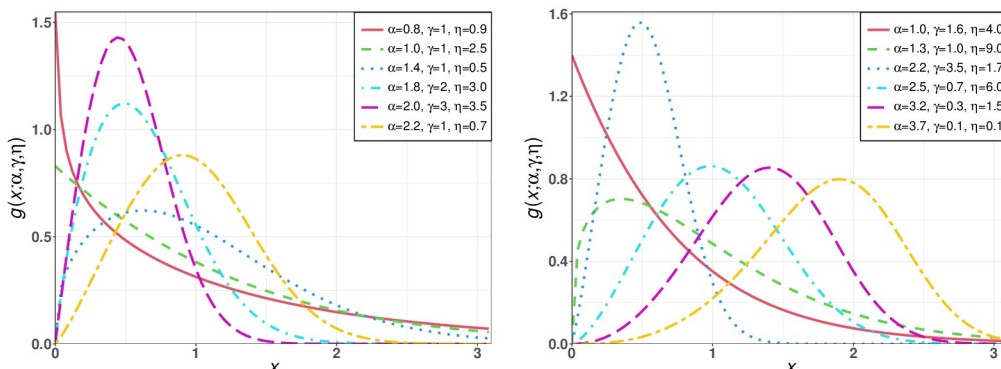

**Fig 1. Some density plots for the TIEx-W distribution.**

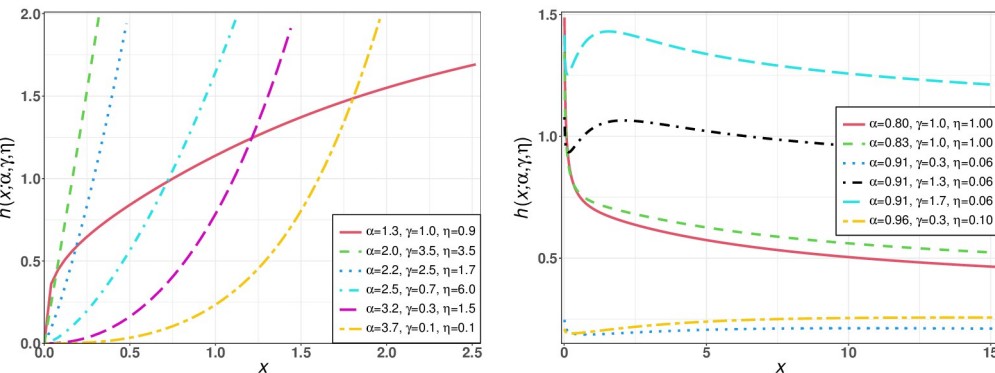

**Fig 2. Some hrf plots for the TIEx-W distribution.**

The survival function and the hrf of the TIEx-W are, respectively, given by

$$S(x; \alpha, \gamma, \eta) = \frac{e - e^{(1-e^{-\gamma x^{\alpha}})} + e^{-\gamma x^{\alpha}}}{e - \bar{\eta}} \tag{8}$$

and

$$h(x; \alpha, \gamma, \eta) = \frac{\alpha \gamma x^{\alpha-1} e^{-\gamma x^{\alpha}} \left(e^{(1-e^{-\gamma x^{\alpha}})} + \eta\right)}{e - e^{(1-e^{-\gamma x^{\alpha}})} + e^{-\gamma x^{\alpha}}}. \tag{9}$$

Fig 2 displays some hrf plots for the TIEx-W distribution. It can be seen that the hrf shapes of this model are: increasing, decreasing and modified bathtub.

## 4 Distributional properties

In this section, we derive some general properties of the TIEx-F family including: quantile function, median, $r^{th}$ moments and moment generating function, shapes of TIEx-W pdf and the order statistics.

### 4.1 The quantile function

The quantile function (qf) of the TIEx-F class follows by inverting the cdf (4). Thus, we have

$$Q(u) = F^{-1}\left[\frac{1}{\eta}\left(1 - u \ (1 - e - \eta) - \eta \ W_0\left(\frac{1}{\eta}e^{\frac{1-u \ (1-e-\eta)}{\eta}}\right)\right)\right], \tag{10}$$

where $W_0(\cdot)$ is the principal branch of the Lambert function, and $u$ follows the uniform distribution (0, 1).

Eq (10) can be adopted to generate random numbers from the TIEx-F family distributions. So, the quantaile function for the TIEx-W distribution can be written as

$$Q(u) = \left[\frac{1}{\gamma}\log\left(-\frac{\eta}{1 - \eta - u \ (1 - e - \eta) - \eta \ W_0\left(\frac{1}{\eta}e^{\frac{1-u \ (1-e-\eta)}{\eta}}\right)}\right)\right]^{1/\alpha}. \tag{11}$$

## 4.2 The median

The median of the TIEx-*F* family distributions can be obtained using $u = \frac{1}{2}$ in (10), and it is given by

$$M = Q\left(\frac{1}{2}\right) = F^{-1}\left[\frac{1}{\eta}\left(1 - \frac{1}{2}(1 - e - \eta) - \eta \ W_0\left(\frac{1}{\eta}e^{\frac{1 - \frac{1}{2}(1-e-\eta)}{\eta}}\right)\right)\right],$$

and of the TIEx-W distribution is given by

$$M = Q\left(\frac{1}{2}\right) = \left[\frac{1}{\gamma}\log\left(-\frac{\eta}{1 - \eta - \frac{1}{2}(1 - e - \eta) - \eta \ W_0\left(\frac{1}{\eta}e^{\frac{1 - \frac{1}{2}(1-e-\eta)}{\eta}}\right)}\right)\right]^{1/\alpha}.$$

## 4.3 The $r^{th}$ moments and moment generating function

Let $X \sim$ TIEx-*F* $(\eta, \Psi)$, hence the $r^{th}$ moment of $X$ takes the form

$$\mu_r' = \int_{-\infty}^{\infty} x^r g(x; \eta, \Psi)dx. \tag{12}$$

By inserting (5) in (12), we obtain

$$\mu_r' = \int_{-\infty}^{\infty} x^r \frac{f(x, \Psi)[e^{F(x, \Psi)} + \eta]}{(e - \bar{\eta})} dx. \tag{13}$$

The Maclaurin series is applied to the exponential function $e^y$ as follows

$$e^y = 1 + x + \frac{x^2}{2!} + \frac{x^3}{3!} + \ldots = \sum_{n=0}^{\infty} \frac{y^n}{n!}. \tag{14}$$

Using $y = F(x; \Psi)$ in (14)

$$e^{F(x;\Psi)} = \sum_{n=0}^{\infty} \frac{F(x; \Psi)^n}{n!}. \tag{15}$$

Hence, the $r^{th}$ moment follows as

$$\mu_r' = \sum_{n=0}^{\infty} \frac{\Lambda_{r,n}}{(e - \bar{\eta})n!} + \frac{\eta \Lambda_r}{(e - \bar{\eta})}, \tag{16}$$

where $\Lambda_{r,n} = \int_{-\infty}^{\infty} x^r f(x, \Psi) \ F(x; \Psi)^n \ dx$, and $\Lambda_r = \int_{-\infty}^{\infty} x^r f(x, \Psi) \ dx$.

The moment generating function of the TIEx-*F* family has the form $M_X(t)$, is given by

$$M_X(t) = \sum_{r,n=0}^{\infty} \frac{t^r \Lambda_{r,n}}{(e - \bar{\eta})r!n!} + \sum_{r=0}^{\infty} \frac{t^r \eta \Lambda_r}{(e - \bar{\eta})r!}.$$

The first four moments of the TIEx-*F* family follows for *r* = 1,2,3,4. Numerical values for the mean, variance, skewness (Sk) and kurtosis (Kur) of the TIEx-W distribution for some

**Table 1. Descriptive measures of TIEx-W distribution for $\alpha = 1.5$, $\gamma = 1$ and several values of $\eta$.**

| $\eta$ | Mean | Variance | Sk | Kur |
|---|---|---|---|---|
| 0.5 | 1.034349 | 0.415879 | 0.886898 | 3.900027 |
| 3 | 0.964618 | 0.398899 | 0.978853 | 4.113748 |
| 6 | 0.940569 | 0.390788 | 1.014008 | 4.211114 |
| 9 | 0.929982 | 0.386850 | 1.029937 | 4.258030 |
| 12 | 0.914026 | 0.324536 | 1.139009 | 4.485534 |
| 15 | 0.900207 | 0.293016 | 1.494864 | 5.30359 |
| 18 | 0.855505 | 0.255408 | 1.678955 | 5.816348 |

parametric values are reported in Tables 1 and 2. These tables show that the new additional parameter provides more flexibility to the TIEx-W distribution in terms of its Sk and Kur.

## 4.4 Shapes of TIEx-W pdf

The behavior of the pdf in (7) when $x \to 0$ and $x \to \infty$ are, respectively, given by

$$\lim_{x \to 0} g(x; \alpha, \gamma, \eta) = \begin{cases} \infty & \text{if } \alpha < 1, \\ \dfrac{\gamma \ (\eta + 1)}{e - (1 - \eta)} & \text{if } \alpha = 1, , \\ 0 & \text{if } \alpha > 1. \end{cases}$$

$$\lim_{x \to \infty} g(x; \alpha, \gamma, \eta) = 0.$$

This clearly appears in Fig 1.

## 4.5 Order statistics

Let $X_1, X_2, \ldots .X_n$ be a random sample from (7) and $X_{1:n} \leq X_{2:n} \leq \ldots \leq X_{n:n}$ denote the the corresponding order statistics. It is well known that the pdf and the cdf of the of $r^{th}$ order statistics, say, $X_{r:n}$ and $1 \leq r \leq n$, respectively, are given by

$$\begin{aligned} g_{r:n}(x; \eta, \boldsymbol{\Psi}) &= \frac{n!}{(r-1)!(n-r)!} [G(x; \eta, \boldsymbol{\Psi})]^{r-1} [1 - G(x; \eta, \boldsymbol{\Psi})]^{n-r} g(x; \eta, \boldsymbol{\Psi}) \\ &= \frac{n!}{(r-1)!(n-r)!} \sum_{u=0}^{n-r} (-1)^u \binom{n-r}{u} [G(x; \eta, \boldsymbol{\Psi})]^{r-1+u} g(x; \eta, \boldsymbol{\Psi}) \end{aligned} \tag{17}$$

**Table 2. Descriptive measures of TIEx-W distribution for $\eta = 1.5$, $\gamma = 1$ and several values of $\alpha$.**

| $\alpha$ | Mean | Variance | Sk | Kur |
|---|---|---|---|---|
| 0.9 | 1.226517 | 1.605750 | 2.106404 | 9.899798 |
| 1.5 | 1.002782 | 0.409397 | 0.926495 | 3.984561 |
| 3 | 0.946664 | 0.106607 | 0.059345 | 2.708730 |
| 6 | 0.956762 | 0.031269 | -0.47328 | 3.175183 |
| 12 | 0.966997 | 0.015069 | -0.68961 | 3.660633 |
| 15 | 0.973592 | 0.008880 | -0.80966 | 4.012897 |
| 18 | 0.978053 | 0.005856 | -0.88636 | 4.266952 |

and

$$
\begin{aligned}
G_{r:n}(x; \eta, \boldsymbol{\Psi}) &= \sum_{l=k}^{n} \binom{n}{l} [G(x; \eta, \boldsymbol{\Psi})]^l [1 - G(x; \eta, \boldsymbol{\Psi})]^{n-l} \\
&= \sum_{l=k}^{n} \sum_{u=0}^{n-r} (-1)^u \binom{n}{l} \binom{n-r}{u} [G(x; \eta, \boldsymbol{\Psi})]^{l+u},
\end{aligned}
\tag{18}
$$

for $k = 1, 2, \ldots, n$. It follows from (17) and (18) that the pdf and cdf of the $r^{th}$ order statistic of the TIEx-F family can be reduced to

$$
\begin{aligned}
g_{r:n}(x; \eta, \boldsymbol{\Psi}) &= \frac{n!}{(r-1)!(n-r)!} \sum_{u=0}^{n-r} (-1)^u \binom{n-r}{u} \left[ \frac{e^{F(x;\boldsymbol{\Psi})} - [1 - \eta F(x; \boldsymbol{\Psi})]}{e - \bar{\eta}} \right]^{r-1+u} \\
&\times \frac{f(x; \boldsymbol{\Psi})[e^{F(x;\boldsymbol{\Psi})} + \eta]}{e - \bar{\eta}}
\end{aligned}
$$

and

$$
G_{r:n}(x; \eta, \boldsymbol{\Psi}) = \sum_{l=k}^{n} \sum_{u=0}^{n-r} (-1)^u \binom{n}{l} \binom{n-r}{u} \left[ \frac{e^{F(x;\boldsymbol{\Psi})} - [1 - \eta F(x; \boldsymbol{\Psi})]}{e - \bar{\eta}} \right]^{l+u}.
$$

So, the pdf and cdf of the $r^{th}$ order statistic of the TIEx-W model can be reduced to

$$
\begin{aligned}
g_{r:n}(x; \alpha, \gamma, \eta) &= \frac{n!}{(r-1)!(n-r)!} \sum_{u=0}^{n-r} (-1)^u \binom{n-r}{u} \\
&\times \left[ \frac{e^{(1-e^{-\gamma x^\alpha})} - [1 - \eta(1 - e^{-\gamma x^\alpha})]}{e - \bar{\eta}} \right]^{r-1+u} \\
&\times \frac{\alpha \gamma x^{\alpha-1} e^{-\gamma x^\alpha} (e^{(1-e^{-\gamma x^\alpha})} + \eta)}{e - \bar{\eta}}
\end{aligned}
$$

and

$$
G_{r:n}(x; \alpha, \gamma, \eta) = \sum_{l=k}^{n} \sum_{u=0}^{n-r} (-1)^u \binom{n}{l} \binom{n-r}{u} \left[ \frac{e^{(1-e^{-\gamma x^\alpha})} - [1 - \eta(1 - e^{-\gamma x^\alpha})]}{e - \bar{\eta}} \right]^{l+u}.
$$

## 5 Estimation for the TIEx-W parameters

In this section, eight estimation methods are considered to estimate the unknown parameters of the TIEx-W model.

### 5.1 Maximum likelihood method

Consider a random sample from the TIEx-W model with pdf given by (7), denoted by $x_1, \ldots, x_m$, and their associated observed order statistics, denoted by $x_{(1)}, x_{(2)}, \cdots, x_{(n)}$. Then, the log-

likelihood function reduces to

$$L(\alpha, \gamma, \eta) = m \log(\alpha \gamma) - m \log(e - \bar{\eta}) - (\alpha - 1) \sum_{k=1}^{m} \log(x_k)$$

$$-\gamma \sum_{k=1}^{m} x_k^{\alpha} + \sum_{k=1}^{m} \log(e^{(1-e^{-\gamma x_k^{\alpha}})} + \eta).$$

(19)

The maximum likelihood estimates (MLEs) of $\alpha$, $\gamma$ and $\eta$ can be determined by maximizing (19) with respect to $\alpha$, $\gamma$ and $\eta$ or by solving the following two non linear equations

$$\frac{\partial L(\alpha, \gamma, \eta)}{\partial \alpha} = \frac{m}{\alpha} - \sum_{k=1}^{m} \log(x_k) - \alpha \gamma \sum_{k=1}^{m} x_k^{\alpha-1} + \sum_{k=1}^{m} \frac{\gamma x_k^{\alpha} e^{(1-e^{-\gamma x_k^{\alpha}})} \log(x_k)}{e^{\gamma x_k^{\alpha}} (e^{(1-e^{-\gamma x_k^{\alpha}})} + \eta)} = 0,$$

(20)

$$\frac{\partial L(\alpha, \gamma, \eta)}{\partial \gamma} = \frac{m}{\gamma} - \sum_{k=1}^{m} x_k^{\alpha} + \sum_{k=1}^{m} \frac{x_k^{\alpha} e^{-\gamma x_k^{\alpha}} e^{(1-e^{-\gamma x_k^{\alpha}})}}{e^{(1-e^{-\gamma x_k^{\alpha}})} + \eta} = 0$$

(21)

and

$$\frac{\partial L(\alpha, \gamma, \eta)}{\partial \eta} = \frac{m}{(\bar{\eta} - e)} + \sum_{k=1}^{m} \frac{1}{e^{(1-e^{-\gamma x_k^{\alpha}})} + \eta} = 0.$$

(22)

The three Eqs (20)–(22) have no explicit solutions, hence numerical techniques will be employed to obtain the MLEs of the parameters.

## 5.2 Maximum product of spacings method

The maximum product of spacings estimates (MPSEs) for $\alpha$, $\gamma$ and $\eta$ of the TIEx-W model can be obtained by maximizing the following function with respect to $\alpha$, $\gamma$ and $\eta$

$$M(\alpha, \gamma, \eta) = \frac{1}{m+1} \sum_{k=1}^{m+1} \log \left\{ \frac{e^{(1-e^{-\gamma x_k^{\alpha}})} - [1 - \eta(1 - e^{-\gamma x_k^{\alpha}})]}{e - \bar{\eta}} \right.$$

$$\left. - \frac{e^{(1-e^{-\gamma x_{k-1}^{\alpha}})} - [1 - \eta(1 - e^{-\gamma x_{k-1}^{\alpha}})]}{e - \bar{\eta}} \right\}.$$

(23)

The MPSEs of the parameters $\alpha$, $\gamma$ and $\eta$, denoted by $\hat{\alpha}_{MP}$, $\hat{\gamma}_{MP}$ and $\hat{\eta}_{MP}$, follows by maximizing Eq (23) or by solving the following three equations simultaneously

$$\frac{\partial M(\alpha, \gamma, \eta)}{\partial \alpha} = \frac{1}{m+1} \sum_{k=1}^{m+1} \frac{\varrho_k(\alpha, \gamma, \eta) - \varrho_{k-1}(\alpha, \gamma, \eta)}{\upsilon_k(\alpha, \gamma, \eta)} = 0,$$

$$\frac{\partial M(\alpha, \gamma, \eta)}{\partial \gamma} = \frac{1}{m+1} \sum_{k=1}^{m+1} \frac{\vartheta_k(\alpha, \gamma, \eta) - \vartheta_{k-1}(\alpha, \gamma, \eta)}{\upsilon_k(\alpha, \gamma, \eta)} = 0$$

and

$$\frac{\partial M(\alpha, \gamma, \eta)}{\partial \eta} = \frac{1}{m+1} \sum_{k=1}^{m+1} \frac{\varphi_k(\alpha, \gamma, \eta) - \varphi_{k-1}(\alpha, \gamma, \eta)}{\upsilon_k(\alpha, \gamma, \eta)} = 0,$$

where

$$v_k(\alpha, \gamma, \eta) = \left\{ \frac{e^{(1-e^{-\gamma x_k^\alpha})} - [1 - \eta(1 - e^{-\gamma x_k^\alpha})]}{e - \bar{\eta}} \right.$$
$$\left. - \frac{e^{(1-e^{-\gamma x_{k-1}^\alpha})} - [1 - \eta(1 - e^{-\gamma x_{k-1}^\alpha})]}{e - \bar{\eta}} \right\},$$

$$\varrho_k(\alpha, \gamma, \eta) = \frac{1}{(e - \bar{\eta})} \gamma x_k^\alpha e^{-\gamma x_k^\alpha} \left[ \eta + e^{(1-e^{-\gamma x_k^\alpha})} \right] \log(x_k), \tag{24}$$

$$\vartheta_k(\alpha, \gamma, \eta) = \frac{1}{(e - \bar{\eta})} x_k^\alpha e^{-\gamma x_k^\alpha} \left[ \eta + e^{(1-e^{-\gamma x_k^\alpha})} \right] \tag{25}$$

and

$$\varphi_k(\alpha, \gamma, \eta) = \frac{1}{(e - \bar{\eta})^4} \left\{ (e - \bar{\eta})^3 (1 - e^{-\gamma x_k^\alpha}) - [1 - \eta(1 - e^{-\gamma x_k^\alpha})] \right\} - \frac{e^{1-e^{-\gamma x_k^\alpha}}}{(e - \bar{\eta})^2}. \tag{26}$$

## 5.3 Least squares and weighted least squares methods

The least squares estimates (LSEs) and weighted least squares estimates (WLSEs) [18] of the TIEx-W parameters can be obtained by minimizing the following function

$$S(\alpha, \gamma, \eta) = \sum_{k=1}^m \delta_k \left\{ \frac{e^{(1-e^{-\gamma x_k^\alpha})} - [1 - \eta(1 - e^{-\gamma x_k^\alpha})]}{e - \bar{\eta}} - \frac{k}{m+1} \right\}^2,$$

where $\delta_k = \frac{(m+2)\,(m+1)^2}{k(m-k+1)}$ for the WLS method and $\delta_k = 1$ for the LS method. Practically, the LSEs of the parameters $\alpha$, $\gamma$ and $\eta$ of the TIEx-W model, denoted by $\hat{\alpha}_{LS}$, $\hat{\gamma}_{Ls}$ and $\hat{\eta}_{LS}$, and the WLSEs, denoted by $\hat{\alpha}_{WL}$, $\hat{\gamma}_{WL}$ and $\hat{\eta}_{WL}$, are obtained by solving the following two equations simultaneously

$$\frac{\partial S(\alpha, \gamma, \eta)}{\partial \alpha} = \sum_{k=1}^m \delta_k \left\{ \frac{e^{(1-e^{-\gamma x_k^\alpha})} - [1 - \eta(1 - e^{-\gamma x_k^\alpha})]}{e - \bar{\eta}} - \frac{k}{m+1} \right\} \varrho_k(\alpha, \gamma, \eta) = 0,$$

$$\frac{\partial S(\alpha, \gamma, \eta)}{\partial \gamma} = \sum_{k=1}^m \delta_k \left\{ \frac{e^{(1-e^{-\gamma x_k^\alpha})} - [1 - \eta(1 - e^{-\gamma x_k^\alpha})]}{e - \bar{\eta}} - \frac{k}{m+1} \right\} \vartheta_k(\alpha, \gamma, \eta) = 0$$

and

$$\frac{\partial S(\alpha, \gamma, \eta)}{\partial \eta} = \sum_{k=1}^m \delta_k \left\{ \frac{e^{(1-e^{-\gamma x_k^\alpha})} - [1 - \eta(1 - e^{-\gamma x_k^\alpha})]}{e - \bar{\eta}} - \frac{k}{m+1} \right\} \varphi_k(\alpha, \gamma, \eta) = 0,$$

where $\varrho_k(\alpha, \gamma, \eta)$, $\vartheta_k(\alpha, \gamma, \eta)$ and $\varphi_k(\alpha, \gamma, \eta)$ are given by (24)–(26).

## 5.4 Cramér-von-Mises and percentiles methods

The Cramér-von Mises estimates (CVMEs) of the parameters of the TIEx-W model, say $\hat{\alpha}_{CM}$ and $\hat{\gamma}_{CM}$, follows by minimizing the following equation with respect to the parameters $\alpha$, $\gamma$ and $\eta$

$$C(\alpha, \gamma, \eta) = \sum_{k=1}^{m} \left\{ \frac{e^{(1-e^{-\gamma x_k^{\alpha}})} - [1 - \eta(1 - e^{-\gamma x_k^{\alpha}})]}{e - \bar{\eta}} - \frac{2k-1}{2m} \right\}^2 . \qquad (27)$$

Equivalently, the CVMEs of the parameters $\alpha$, $\gamma$ and $\eta$ are obtained by solving the following two equations

$$\frac{\partial C(\alpha, \gamma, \eta)}{\partial \alpha} = \sum_{k=1}^{m} \left\{ \frac{e^{(1-e^{-\gamma x_k^{\alpha}})} - [1 - \eta(1 - e^{-\gamma x_k^{\alpha}})]}{e - \bar{\eta}} - \frac{2k-1}{2m} \right\} \varrho_k(\alpha, \gamma, \eta) = 0,$$

$$\frac{\partial C(\alpha, \gamma, \eta)}{\partial \gamma} = \sum_{k=1}^{m} \left\{ \frac{e^{(1-e^{-\gamma x_k^{\alpha}})} - [1 - \eta(1 - e^{-\gamma x_k^{\alpha}})]}{e - \bar{\eta}} - \frac{2k-1}{2m} \right\} \vartheta_k(\alpha, \gamma, \eta) = 0$$

and

$$\frac{\partial C(\alpha, \gamma, \eta)}{\partial \eta} = \sum_{k=1}^{m} \left\{ \frac{e^{(1-e^{-\gamma x_k^{\alpha}})} - [1 - \eta(1 - e^{-\gamma x_k^{\alpha}})]}{e - \bar{\eta}} - \frac{2k-1}{2m} \right\} \varphi_k(\alpha, \gamma, \eta) = 0,$$

$\varrho_k(\alpha, \gamma, \eta)$, $\vartheta_k(\alpha, \gamma, \eta)$ and $\varphi_k(\alpha, \gamma, \eta)$ are given by (24)–(26).

The percentiles estimation method was proposed by [19, 20]. The percentiles estimates (PCEs) of the TIEx-W parameters $\alpha$, $\gamma$ and $\eta$, denoted by $\hat{\alpha}_{PE}$, $\hat{\gamma}_{PE}$ and $\hat{\eta}_{PE}$, can be obtained by minimizing

$$P(\alpha, \gamma, \eta) = \sum_{k=1}^{m} \{x_{(k)} - Q(u)\}^2, \qquad (28)$$

where $Q(u)$ denotes the quantile function of the TIEx-X distribution and it has no closed form expression. Then, numerical techniques are employed to generate data from the TIEx-X distribution.

## 5.5 Anderson-Darling and right-tail Anderson-Darling methods

The Anderson-Darling (AD) method is known as a type of minimum distance estimators which can be obtained by minimizing the AD statistic. For the TIEx-W model, the AD estimates (ADEs) of the TIEx-W parameters $\alpha$, $\gamma$ and $\eta$, say $\hat{\alpha}_{AD}$, $\hat{\gamma}_{AD}$ and $\hat{\eta}_{AD}$, can be obtained by minimizing

$$AD(\alpha, \gamma, \eta) = -m - \frac{1}{m} \sum_{k=1}^{m} (2k-1)[\log(F(x_{(k)}|\alpha, \gamma, \eta)) + \log(\bar{F}(x_{(m-k+1)}|\alpha, \gamma, \eta))],$$

with respect to $\alpha$, $\gamma$ and $\eta$, where $\bar{F}(.) = 1 - F(.)$. These estimates can also be obtained by

solving the following equations

$$\frac{\partial AD(\alpha, \gamma, \eta)}{\partial \alpha} = \sum_{k=1}^{m} (2k-1) \frac{\varrho_k(\alpha, \gamma, \eta)}{F(x_{(k)}|\alpha, \gamma, \eta)} - \sum_{k=1}^{m} (2k-1) \frac{\varrho_{m-k+1}(\alpha, \gamma, \eta)}{1 - F(x_{(m-k+1)}|\alpha, \gamma, \eta)} = 0,$$

$$\frac{\partial AD(\alpha, \gamma, \eta)}{\partial \gamma} = \sum_{k=1}^{m} (2k-1) \frac{\vartheta_k(\alpha, \gamma, \eta)}{F(x_{(k)}|\alpha, \gamma, \eta)} - \sum_{k=1}^{m} (2k-1) \frac{\vartheta_{m-k+1}(\alpha, \gamma, \eta)}{1 - F(x_{(m-k+1)}|\alpha, \gamma, \eta)} = 0$$

and

$$\frac{\partial AD(\alpha, \gamma, \eta)}{\partial \gamma} = \sum_{k=1}^{m} (2k-1) \frac{\varphi_k(\alpha, \gamma, \eta)}{F(x_{(k)}|\alpha, \gamma, \eta)} - \sum_{k=1}^{m} (2k-1) \frac{\varphi_{m-k+1}(\alpha, \gamma, \eta)}{1 - F(x_{(m-k+1)}|\alpha, \gamma, \eta)} = 0,$$

where $\varrho_k(\alpha, \gamma, \eta)$, $\vartheta_k(\alpha, \gamma, \eta)$ and $\varphi_k(\alpha, \gamma, \eta)$ are given by (24)–(26).

Similarly, the right-tail Anderson-Darling estimates (RADEs) of the TIEx-W parameters $\alpha$, $\gamma$ and $\eta$, say $\hat{\alpha}_{RAD}$, $\hat{\gamma}_{RAD}$ and $\hat{\eta}_{RAD}$, are obtained by minimizing

$$RAD(\alpha, \gamma, \eta) = \frac{m}{2} - 2 \sum_{k=1}^{n} F\left(x_{(k)} \mid \alpha, \gamma, \eta\right) - \frac{1}{m} \sum_{k=1}^{n} (2k-1) \log \bar{F}\left(x_{(m-k+1)} \mid \alpha, \gamma, \eta\right),$$

with respect to $\alpha$, $\gamma$ and $\eta$. Furthermore, these estimates can also be obtained by solving the following equations simultaneously

$$\frac{\partial RAD(\alpha, \gamma, \eta)}{\partial \alpha} = -2 \sum_{k=1}^{m} \varrho_k(\alpha, \gamma, \eta) + \frac{1}{m} \sum_{k=1}^{m} (2k-1) \frac{\varrho_{m-k+1}(\alpha, \gamma, \eta)}{1 - F(x_{(m-k+1)}|\alpha, \gamma, \eta)} = 0,$$

$$\frac{\partial RAD(\alpha, \gamma, \eta)}{\partial \gamma} = -2 \sum_{k=1}^{m} \vartheta_k(\alpha, \gamma, \eta) + \frac{1}{m} \sum_{k=1}^{m} (2k-1) \frac{\vartheta_{m-k+1}(\alpha, \gamma, \eta)}{1 - F(x_{(m-k+1)}|\alpha, \gamma, \eta)} = 0$$

and

$$\frac{\partial RAD(\alpha, \gamma, \eta)}{\partial \eta} = -2 \sum_{k=1}^{m} \varphi_k(\alpha, \gamma, \eta) + \frac{1}{m} \sum_{k=1}^{m} (2k-1) \frac{\varphi_{m-k+1}(\alpha, \gamma, \eta)}{1 - F(x_{(m-k+1)}|\alpha, \gamma, \eta)} = 0,$$

where $\varrho_k(\alpha, \gamma, \eta)$, $\vartheta_k(\alpha, \gamma, \eta)$ and $\varphi_k(\alpha, \gamma, \eta)$ are given by (24)–(26).

## 6 Simulation results

In this section, we have carried out an extensive simulation study to assess and compare the performance of the eight frequentist estimators. The methods are explored for $n = \{20, 50, 100, 200, 400\}$ with parameter values $\alpha = (2.75)$, $\gamma = (0.5, 2.0)$ and $\eta = (0.67, 1.5)$. We generate $N = 5000$ random samples from the TIEx-W distribution using the inverse transform method. The following procedures are adopted to generate the data from the TIEx-W distribution:

- Step 1: Generate random values from the TIEx-W distribution with size $n$.

- Step 2: Using the obtained samples in step 1, calculate $\hat{\alpha}$, $\hat{\gamma}$ and $\hat{\eta}$ via 1-MLES, 2-MPSEs, 3-LSEs, 4-CVMEs, 5-WLSEs, 6-PCEs, 7-ADEs, 8-RADEs.

- Step 3: Repeat the steps 1 and 2, $N$ times.

For each estimate, we calculate average estimates of the parameters (AVEs), mean-squared errors (MSEs), absolute bias (ABs), and mean relative errors (MREs). The formulae of these measures are given for $\boldsymbol{\phi} = (\alpha, \gamma, \eta)^{\top}$ by

**Table 3. Simulation results of the eight estimation methods for $\phi = (\alpha = 2.75, \gamma = 2.00, \eta = 0.67)^{\top}$.**

| n | Measures | Est. Par. | MLEs | MPSEs | LSEs | CVMEs | WLSEs | PCEs | ADEs | RADEs |
|---|---|---|---|---|---|---|---|---|---|---|
| 20 | AVEs | $\hat{\alpha}$ | 2.75624 | 2.35767 | 2.39565 | 2.64250 | 2.40206 | 2.42096 | 2.62632 | 2.77081 |
| | | $\hat{\gamma}$ | 1.72155 | 2.19706 | 1.99302 | 1.72319 | 2.24302 | 2.05615 | 1.89511 | 2.16619 |
| | | $\hat{\eta}$ | 0.74185 | 0.59915 | 0.64737 | 0.73176 | 0.59813 | 0.63529 | 0.69099 | 0.64290 |
| | MSEs | $\hat{\alpha}$ | 1.44036 | 1.39733 | 2.13957 | 2.34009 | 1.41448 | 1.79126 | 1.46813 | 2.40870 |
| | | $\hat{\gamma}$ | 2.85766 | 3.17883 | 3.82296 | 3.73225 | 3.31452 | 3.60669 | 3.10305 | 3.68172 |
| | | $\hat{\eta}$ | 0.21730 | 0.22421 | 0.34959 | 0.33569 | 0.26117 | 0.32641 | 0.25616 | 0.32126 |
| | ABs | $\hat{\alpha}$ | 1.20015 | 1.18209 | 1.46273 | 1.52974 | 1.18932 | 1.33838 | 1.21166 | 1.55200 |
| | | $\hat{\gamma}$ | 1.69046 | 1.78293 | 1.95524 | 1.93190 | 1.82058 | 1.89913 | 1.76155 | 1.91878 |
| | | $\hat{\eta}$ | 0.46616 | 0.47351 | 0.59126 | 0.57939 | 0.51105 | 0.57132 | 0.50612 | 0.56680 |
| | MREs | $\hat{\alpha}$ | 0.43642 | 0.42985 | 0.53190 | 0.55627 | 0.43248 | 0.48668 | 0.44060 | 0.56436 |
| | | $\hat{\gamma}$ | 0.84523 | 0.89146 | 0.97762 | 0.96595 | 0.91029 | 0.94956 | 0.88077 | 0.95939 |
| | | $\hat{\eta}$ | 0.69576 | 0.70673 | 0.88248 | 0.86476 | 0.76275 | 0.85272 | 0.75540 | 0.84597 |
| 50 | AVEs | $\hat{\alpha}$ | 2.76195 | 2.51121 | 2.58904 | 2.79208 | 2.68955 | 2.61937 | 2.72290 | 2.79844 |
| | | $\hat{\gamma}$ | 1.91971 | 2.12357 | 1.92990 | 2.12100 | 2.27972 | 1.94300 | 1.96959 | 2.17193 |
| | | $\hat{\eta}$ | 0.69372 | 0.62512 | 0.67348 | 0.65466 | 0.60808 | 0.67164 | 0.67847 | 0.64505 |
| | MSEs | $\hat{\alpha}$ | 0.57487 | 0.55864 | 1.07358 | 1.12896 | 0.64893 | 0.77387 | 0.66492 | 1.21394 |
| | | $\hat{\gamma}$ | 1.48154 | 1.54704 | 2.57226 | 2.53167 | 1.89094 | 1.93310 | 1.67940 | 2.33034 |
| | | $\hat{\eta}$ | 0.07457 | 0.07332 | 0.16618 | 0.16265 | 0.09270 | 0.10313 | 0.08899 | 0.11982 |
| | ABs | $\hat{\alpha}$ | 0.75820 | 0.74742 | 1.03614 | 1.06253 | 0.80556 | 0.87970 | 0.81543 | 1.10179 |
| | | $\hat{\gamma}$ | 1.21719 | 1.24380 | 1.60383 | 1.59112 | 1.37512 | 1.39036 | 1.29592 | 1.52654 |
| | | $\hat{\eta}$ | 0.27307 | 0.27077 | 0.40766 | 0.40330 | 0.30447 | 0.32114 | 0.29830 | 0.34616 |
| | MREs | $\hat{\alpha}$ | 0.27571 | 0.27179 | 0.37678 | 0.38637 | 0.29293 | 0.31989 | 0.29652 | 0.40065 |
| | | $\hat{\gamma}$ | 0.60859 | 0.62190 | 0.80191 | 0.79556 | 0.68756 | 0.69518 | 0.64796 | 0.76327 |
| | | $\hat{\eta}$ | 0.40756 | 0.40414 | 0.60844 | 0.60194 | 0.45443 | 0.47932 | 0.44523 | 0.51665 |
| 100 | AVEs | $\hat{\alpha}$ | 2.75987 | 2.62363 | 2.70171 | 2.70120 | 2.76422 | 2.68455 | 2.73279 | 2.81653 |
| | | $\hat{\gamma}$ | 1.98141 | 2.08244 | 2.07212 | 1.94446 | 2.27257 | 1.95607 | 1.98239 | 2.11577 |
| | | $\hat{\eta}$ | 0.67592 | 0.63876 | 0.65424 | 0.68058 | 0.61564 | 0.67465 | 0.67247 | 0.65215 |
| | MSEs | $\hat{\alpha}$ | 0.26759 | 0.25780 | 0.58047 | 0.54830 | 0.32075 | 0.37360 | 0.35242 | 0.57912 |
| | | $\hat{\gamma}$ | 0.75494 | 0.79775 | 1.60433 | 1.52309 | 1.03080 | 1.10899 | 0.94792 | 1.35441 |
| | | $\hat{\eta}$ | 0.03084 | 0.03008 | 0.07602 | 0.07596 | 0.04202 | 0.05023 | 0.04228 | 0.05593 |
| | ABs | $\hat{\alpha}$ | 0.51729 | 0.50774 | 0.76189 | 0.74048 | 0.56635 | 0.61123 | 0.59365 | 0.76100 |
| | | $\hat{\gamma}$ | 0.86887 | 0.89317 | 1.26662 | 1.23414 | 1.01528 | 1.05308 | 0.97361 | 1.16379 |
| | | $\hat{\eta}$ | 0.17561 | 0.17344 | 0.27572 | 0.27561 | 0.20498 | 0.22413 | 0.20563 | 0.23649 |
| | MREs | $\hat{\alpha}$ | 0.18811 | 0.18463 | 0.27705 | 0.26926 | 0.20594 | 0.22226 | 0.21587 | 0.27673 |
| | | $\hat{\gamma}$ | 0.43444 | 0.44658 | 0.63331 | 0.61707 | 0.50764 | 0.52654 | 0.48680 | 0.58190 |
| | | $\hat{\eta}$ | 0.26210 | 0.25887 | 0.41153 | 0.41136 | 0.30594 | 0.33452 | 0.30691 | 0.35297 |
| 200 | AVEs | $\hat{\alpha}$ | 2.75079 | 2.66858 | 2.72422 | 2.75894 | 2.76600 | 2.71390 | 2.74620 | 2.78890 |
| | | $\hat{\gamma}$ | 1.96534 | 2.04181 | 1.97628 | 2.02185 | 2.15149 | 1.98707 | 2.01250 | 2.06069 |
| | | $\hat{\eta}$ | 0.67979 | 0.65299 | 0.67278 | 0.66696 | 0.63767 | 0.67100 | 0.66785 | 0.65997 |
| | MSEs | $\hat{\alpha}$ | 0.13080 | 0.11065 | 0.29241 | 0.29518 | 0.16193 | 0.17075 | 0.17373 | 0.29994 |
| | | $\hat{\gamma}$ | 0.40931 | 0.34939 | 0.84520 | 0.88509 | 0.49341 | 0.52890 | 0.51518 | 0.70687 |
| | | $\hat{\eta}$ | 0.01637 | 0.01194 | 0.03816 | 0.03816 | 0.01861 | 0.02122 | 0.02068 | 0.02692 |
| | ABs | $\hat{\alpha}$ | 0.36166 | 0.33265 | 0.54075 | 0.54331 | 0.40240 | 0.41322 | 0.41681 | 0.54767 |
| | | $\hat{\gamma}$ | 0.63977 | 0.59109 | 0.91935 | 0.94079 | 0.70243 | 0.72725 | 0.71776 | 0.84076 |
| | | $\hat{\eta}$ | 0.12794 | 0.10929 | 0.19534 | 0.19536 | 0.13640 | 0.14569 | 0.14380 | 0.16408 |
| | MREs | $\hat{\alpha}$ | 0.13151 | 0.12096 | 0.19664 | 0.19757 | 0.14633 | 0.15026 | 0.15157 | 0.19915 |
| | | $\hat{\gamma}$ | 0.31989 | 0.29554 | 0.45967 | 0.47040 | 0.35122 | 0.36363 | 0.35888 | 0.42038 |
| | | $\hat{\eta}$ | 0.19096 | 0.16312 | 0.29155 | 0.29158 | 0.20359 | 0.21744 | 0.21463 | 0.24490 |

(*Continued*)

**Table 3.** (Continued)

| n | Measures | Est. Par. | MLEs | MPSEs | LSEs | CVMEs | WLSEs | PCEs | ADEs | RADEs |
|---|----------|-----------|------|-------|------|-------|-------|------|------|-------|
| 400 | AVEs | $\hat{\alpha}$ | 2.74732 | 2.70582 | 2.73334 | 2.75649 | 2.77728 | 2.72239 | 2.76015 | 2.75716 |
| | | $\hat{\gamma}$ | 1.99043 | 2.01837 | 2.01833 | 2.00434 | 2.12210 | 2.00064 | 2.02093 | 2.01438 |
| | | $\hat{\eta}$ | 0.67117 | 0.65957 | 0.66592 | 0.67495 | 0.64610 | 0.67013 | 0.66668 | 0.66763 |
| | MSEs | $\hat{\alpha}$ | 0.06783 | 0.03960 | 0.14594 | 0.13372 | 0.07942 | 0.08860 | 0.08245 | 0.15177 |
| | | $\hat{\gamma}$ | 0.20478 | 0.01276 | 0.50480 | 0.43692 | 0.24577 | 0.28979 | 0.28083 | 0.37956 |
| | | $\hat{\eta}$ | 0.00753 | 0.00173 | 0.01906 | 0.01747 | 0.00928 | 0.01104 | 0.01091 | 0.01383 |
| | ABs | $\hat{\alpha}$ | 0.26044 | 0.19900 | 0.38202 | 0.36568 | 0.28182 | 0.29765 | 0.28715 | 0.38958 |
| | | $\hat{\gamma}$ | 0.45253 | 0.11295 | 0.71049 | 0.66100 | 0.49575 | 0.53832 | 0.52994 | 0.61609 |
| | | $\hat{\eta}$ | 0.08680 | 0.04165 | 0.13806 | 0.13217 | 0.09634 | 0.10506 | 0.10447 | 0.11760 |
| | MREs | $\hat{\alpha}$ | 0.09471 | 0.07236 | 0.13892 | 0.13298 | 0.10248 | 0.10824 | 0.10442 | 0.14167 |
| | | $\hat{\gamma}$ | 0.22627 | 0.05648 | 0.35525 | 0.33050 | 0.24787 | 0.26916 | 0.26497 | 0.30804 |
| | | $\hat{\eta}$ | 0.12956 | 0.06216 | 0.20606 | 0.19727 | 0.14379 | 0.15680 | 0.15592 | 0.17552 |

$MSEs = \frac{1}{N}\sum_{i=1}^{N}(\hat{\boldsymbol{\phi}} - \boldsymbol{\phi})^2$, ABs, $|ABs(\hat{\boldsymbol{\phi}})| = \frac{1}{N}\sum_{i=1}^{N}|\hat{\boldsymbol{\phi}} - \boldsymbol{\phi}|$, and MREs, $MREs = \frac{1}{N}\sum_{i=1}^{N}|\hat{\boldsymbol{\phi}} - \boldsymbol{\phi}|/\boldsymbol{\phi}$.

The performance of the considered estimators are evaluated in terms of absolute bias, mean-squared error and mean relative error. Considering this approach, the most efficient estimation method will be the one whose MREs value is closer to one and bias closer to zero. All simulations are conducted via R software.

In Tables 3–5 we report the values of AVEs, MSEs, ABs, and MREs for the WLSEs, LSEs, MLEs, MPSEs, CVMEs, ADEs, RADEs and PCEs. The results show that all estimators reveal the property of consistency, where the MSEs and MREs decrease as sample size increases, for all parameter combinations. All estimators show the property of consistency for all parameter combinations. In summary, we conclude that the maximum likelihood (ML) method outperforms all other estimation methods. Therefore, ML methods is considered the optimal method for estimating the TIEx-W parameters.

# 7 Modeling insurance data

In this section, we illustrate the applicability and superiority of the TIEx-W distribution by comparing its goodness of fit to other well-known distributions which are used before for modeling financial and insurance data sets.

The analyzed insurance data represents hospital cost in the state of Wisconsin provided by the Office of the Health Care Information, Wisconsin's Department of Health and Human Resources. The data is available at https://github.com/almof1hm/Hospital-costs-in-the-state-of-Wisconsin/blob/main/Data.xlsx.

We compare the fits of the proposed model (TIEx-W) among some competitive distributions, namely: Weibull (Wi) [21], Lindley–Weibull distribution (LiW) [22], exponentiated Weibull (EW) [23], Kumaraswamy–Weibull (KwW) [24], $\alpha$ logarithmic transformed Weibull (ALW) [25], Burr (Bur) [26], Lomax (Lo) [27] and beta Weibull (BW) [28] models, and the special cases Weibull models derived from families defined in (1)–(3), for the insurance data set.

Table 6 lists the MLEs and the corresponding standard errors (SEs) in parentheses of the parameters for all fitted models, and the Kolmogorov-Smirnov (K-S) statistics and $p$-values for the insurance data set. Since the TIEx-W model has the lowest K-S values and the largest $p$-

**Table 4. Simulation results of the eight estimation methods for $\phi = (\alpha = 2.75, \gamma = 0.50, \eta = 0.67)^{\top}$.**

| n | Measures | Est. Par. | MLEs | MPSEs | LSEs | CVMEs | WLSEs | PCEs | ADEs | RADEs |
|---|---|---|---|---|---|---|---|---|---|---|
| 20 | AVEs | $\hat{\alpha}$ | 2.91188 | 2.33929 | 2.47312 | 2.71757 | 2.51818 | 2.35272 | 2.63180 | 3.05570 |
| | | $\hat{\gamma}$ | 0.48842 | 0.50964 | 0.49030 | 0.48550 | 0.53317 | 0.50091 | 0.46894 | 0.60391 |
| | | $\hat{\eta}$ | 0.68486 | 0.64113 | 0.65142 | 0.67555 | 0.63721 | 0.64775 | 0.68818 | 0.63010 |
| | MSEs | $\hat{\alpha}$ | 2.37251 | 2.02382 | 3.39494 | 3.52646 | 2.64139 | 2.26891 | 2.13571 | 3.82407 |
| | | $\hat{\gamma}$ | 0.21315 | 0.20787 | 0.24673 | 0.24729 | 0.23629 | 0.21802 | 0.21021 | 0.24687 |
| | | $\hat{\eta}$ | 0.10056 | 0.10118 | 0.21452 | 0.20433 | 0.14778 | 0.12048 | 0.10929 | 0.16860 |
| | ABs | $\hat{\alpha}$ | 1.54030 | 1.42261 | 1.84254 | 1.87789 | 1.62523 | 1.50629 | 1.46141 | 1.95552 |
| | | $\hat{\gamma}$ | 0.46168 | 0.45593 | 0.49672 | 0.49728 | 0.48610 | 0.46693 | 0.45849 | 0.49686 |
| | | $\hat{\eta}$ | 0.31711 | 0.31809 | 0.46316 | 0.45203 | 0.38442 | 0.34710 | 0.33058 | 0.41061 |
| | MREs | $\hat{\alpha}$ | 0.56011 | 0.51731 | 0.67001 | 0.68287 | 0.59099 | 0.54774 | 0.53142 | 0.71110 |
| | | $\hat{\gamma}$ | 0.92337 | 0.91185 | 0.99345 | 0.99457 | 0.97219 | 0.93386 | 0.91697 | 0.99371 |
| | | $\hat{\eta}$ | 0.47330 | 0.47476 | 0.69128 | 0.67466 | 0.57377 | 0.51807 | 0.49341 | 0.61285 |
| 50 | AVEs | $\hat{\alpha}$ | 2.76145 | 2.56168 | 2.63338 | 2.70804 | 2.69128 | 2.53131 | 2.72030 | 2.88928 |
| | | $\hat{\gamma}$ | 0.48356 | 0.52257 | 0.49526 | 0.47591 | 0.51304 | 0.49393 | 0.48685 | 0.54830 |
| | | $\hat{\eta}$ | 0.67916 | 0.64465 | 0.66333 | 0.68186 | 0.65888 | 0.66002 | 0.68013 | 0.64884 |
| | MSEs | $\hat{\alpha}$ | 0.76152 | 0.76832 | 1.53361 | 1.54728 | 1.14839 | 0.86837 | 0.95346 | 1.70764 |
| | | $\hat{\gamma}$ | 0.09750 | 0.10075 | 0.17144 | 0.17316 | 0.13807 | 0.11248 | 0.12034 | 0.16009 |
| | | $\hat{\eta}$ | 0.03074 | 0.03083 | 0.06862 | 0.07473 | 0.04757 | 0.03707 | 0.03901 | 0.05735 |
| | ABs | $\hat{\alpha}$ | 0.87265 | 0.87654 | 1.23839 | 1.24390 | 1.07163 | 0.93186 | 0.97645 | 1.30677 |
| | | $\hat{\gamma}$ | 0.31225 | 0.31741 | 0.41406 | 0.41613 | 0.37157 | 0.33538 | 0.34690 | 0.40011 |
| | | $\hat{\eta}$ | 0.17533 | 0.17557 | 0.26195 | 0.27337 | 0.21811 | 0.19253 | 0.19752 | 0.23949 |
| | MREs | $\hat{\alpha}$ | 0.31733 | 0.31874 | 0.45032 | 0.45233 | 0.38968 | 0.33886 | 0.35507 | 0.47519 |
| | | $\hat{\gamma}$ | 0.62449 | 0.63483 | 0.82811 | 0.83225 | 0.74314 | 0.67077 | 0.69379 | 0.80022 |
| | | $\hat{\eta}$ | 0.26168 | 0.26205 | 0.39097 | 0.40802 | 0.32554 | 0.28737 | 0.29480 | 0.35744 |
| 100 | AVEs | $\hat{\alpha}$ | 2.77570 | 2.69832 | 2.67411 | 2.77855 | 2.69000 | 2.63432 | 2.72432 | 2.83505 |
| | | $\hat{\gamma}$ | 0.49933 | 0.52083 | 0.50201 | 0.51091 | 0.49836 | 0.50365 | 0.49589 | 0.52854 |
| | | $\hat{\eta}$ | 0.67041 | 0.65280 | 0.66604 | 0.66777 | 0.66851 | 0.66146 | 0.67196 | 0.65873 |
| | MSEs | $\hat{\alpha}$ | 0.39484 | 0.39191 | 0.77864 | 0.88865 | 0.53411 | 0.41497 | 0.50560 | 0.84537 |
| | | $\hat{\gamma}$ | 0.05174 | 0.05438 | 0.10246 | 0.11072 | 0.07282 | 0.05870 | 0.06881 | 0.09371 |
| | | $\hat{\eta}$ | 0.01395 | 0.01513 | 0.03251 | 0.03524 | 0.02153 | 0.01641 | 0.01940 | 0.02594 |
| | ABs | $\hat{\alpha}$ | 0.62836 | 0.62603 | 0.88241 | 0.94268 | 0.73083 | 0.64418 | 0.71105 | 0.91944 |
| | | $\hat{\gamma}$ | 0.22747 | 0.23319 | 0.32010 | 0.33275 | 0.26986 | 0.24228 | 0.26232 | 0.30613 |
| | | $\hat{\eta}$ | 0.11812 | 0.12300 | 0.18030 | 0.18774 | 0.14675 | 0.12811 | 0.13928 | 0.16106 |
| | MREs | $\hat{\alpha}$ | 0.22849 | 0.22765 | 0.32088 | .34279 | 0.26575 | 0.23425 | 0.25856 | 0.33434 |
| | | $\hat{\gamma}$ | 0.45494 | 0.46638 | 0.64019 | 0.66549 | 0.53971 | 0.48456 | 0.52464 | 0.61226 |
| | | $\hat{\eta}$ | 0.17630 | 0.18358 | 0.26910 | 0.28020 | 0.21903 | 0.19121 | 0.20787 | 0.24039 |
| 200 | AVEs | $\hat{\alpha}$ | 2.75642 | 2.72358 | 2.73158 | 2.73308 | 2.73023 | 2.70604 | 2.73049 | 2.82219 |
| | | $\hat{\gamma}$ | 0.49335 | 0.50458 | 0.50950 | 0.48942 | 0.50161 | 0.50812 | 0.49291 | 0.52547 |
| | | $\hat{\eta}$ | 0.67401 | 0.66296 | 0.66417 | 0.67759 | 0.66760 | 0.66060 | 0.67270 | 0.65772 |
| | MSEs | $\hat{\alpha}$ | 0.18074 | 0.18477 | 0.42040 | 0.42024 | 0.26755 | 0.19801 | 0.24286 | 0.44068 |
| | | $\hat{\gamma}$ | 0.02579 | 0.02403 | 0.05740 | 0.05471 | 0.03968 | 0.02816 | 0.03420 | 0.04893 |
| | | $\hat{\eta}$ | 0.00680 | 0.00656 | 0.01626 | 0.01616 | 0.01043 | 0.00747 | 0.00930 | 0.01217 |
| | ABs | $\hat{\alpha}$ | 0.42513 | 0.42985 | 0.64838 | 0.64826 | 0.51725 | 0.44499 | 0.49281 | 0.66384 |
| | | $\hat{\gamma}$ | 0.16058 | 0.15500 | 0.23958 | 0.23390 | 0.19920 | 0.16782 | 0.18493 | 0.22119 |
| | | $\hat{\eta}$ | 0.08248 | 0.08101 | 0.12750 | 0.12712 | 0.10215 | 0.08642 | 0.09646 | 0.11030 |
| | MREs | $\hat{\alpha}$ | 0.15459 | 0.15631 | 0.23578 | 0.23573 | 0.18809 | 0.16181 | 0.17920 | 0.24140 |
| | | $\hat{\gamma}$ | 0.32116 | 0.31000 | 0.47916 | 0.46781 | 0.39840 | 0.33564 | 0.36987 | 0.44238 |
| | | $\hat{\eta}$ | 0.12311 | 0.12092 | 0.19030 | 0.18973 | 0.15246 | 0.12898 | 0.14397 | 0.16463 |

*(Continued)*

**Table 4.** (Continued)

| n | Measures | Est. Par. | MLEs | MPSEs | LSEs | CVMEs | WLSEs | PCEs | ADEs | RADEs |
|---|---|---|---|---|---|---|---|---|---|---|
| 400 | AVEs | $\hat{\alpha}$ | 2.74072 | 2.73846 | 2.72275 | 2.75357 | 2.73494 | 2.71831 | 2.75870 | 2.78918 |
| | | $\hat{\gamma}$ | 0.49306 | 0.50492 | 0.49426 | 0.49675 | 0.49279 | 0.50363 | 0.50195 | 0.51476 |
| | | $\hat{\eta}$ | 0.67526 | 0.66414 | 0.67248 | 0.67189 | 0.67232 | 0.66654 | 0.66959 | 0.66299 |
| | MSEs | $\hat{\alpha}$ | 0.09496 | 0.08605 | 0.23124 | 0.22958 | 0.13180 | 0.10080 | 0.12672 | 0.21980 |
| | | $\hat{\gamma}$ | 0.01368 | 0.00980 | 0.03201 | 0.03110 | 0.01815 | 0.01434 | 0.01825 | 0.02609 |
| | | $\hat{\eta}$ | 0.00362 | 0.00267 | 0.00831 | 0.00819 | 0.00468 | 0.00361 | 0.00459 | 0.00619 |
| | ABs | $\hat{\alpha}$ | 0.30815 | 0.29334 | 0.48087 | 0.47914 | 0.36304 | 0.31749 | 0.35598 | 0.46883 |
| | | $\hat{\gamma}$ | 0.11694 | 0.09897 | 0.17891 | 0.17635 | 0.13474 | 0.11974 | 0.13508 | 0.16152 |
| | | $\hat{\eta}$ | 0.06013 | 0.05171 | 0.09118 | 0.09047 | 0.06843 | 0.06010 | 0.06772 | 0.07867 |
| | MREs | $\hat{\alpha}$ | 0.11205 | 0.10667 | 0.17486 | 0.17423 | 0.13201 | 0.11545 | 0.12945 | 0.17048 |
| | | $\hat{\gamma}$ | 0.23389 | 0.19794 | 0.35783 | 0.35270 | 0.26948 | 0.23949 | 0.27015 | 0.32304 |
| | | $\hat{\eta}$ | 0.08974 | 0.07718 | 0.13609 | 0.13504 | 0.10213 | 0.08971 | 0.10108 | 0.11742 |

values among all fitted models, we can say the TIEx-W model is best model to the analyzed data set.

The fitted cdf and Kaplan–Meier survival (KMS) plots are displayed in Fig 3. The PP and box plots are sketched in Fig 4. The plots show that the insurance data has a heavier tail and the TIEx-W distribution fits it very closely.

## 8 Bayesian estimation from insurance data

In this section, the insurance data is analyzed using the Bayesian analysis. Let the parameters $\alpha$, $\gamma$ and $\eta$ of TIEx-W distribution have independent gamma priors as

$$\alpha \sim Gamma(a, b), \gamma \sim Gamma(c, d), \eta \sim Gamma(e, f),$$

where $a$, $b$, $c$, $d$, $e$ and $f$ are positive. Then, the joint prior density follows as

$$\pi(\alpha, \gamma, \eta) = \frac{b^a d^c f^e}{\Gamma(a)\Gamma(c)\Gamma(e)} \alpha^{a-1} \gamma^{c-1} \eta^{e-1} e^{-(b\alpha + d\gamma + f\eta)}. \tag{29}$$

We will adopt the well-known five loss functions (LFs) which are listed Table 7, with their Bayes estimators and posterior risk. More information can be explored in [29].

Now, we derive the posterior probability distribution for a complete data. We define the function $\varphi$

$$\varphi(\alpha, \gamma, \eta) = \alpha^{a-1} \gamma^{c-1} \eta^{c-1} e^{-(b\alpha + d\gamma + f\eta)}, \quad \alpha, \gamma, \eta > 0.$$

The joint posterior distribution has the form

$$\pi^*(\alpha, \gamma, \eta | data) \propto \pi(\alpha, \gamma, \eta) L(data). \tag{30}$$

Therefore, the joint posterior density (JPD) of the parameters $\alpha$, $\gamma$ and $\eta$ for complete data is obtained by combining Eq 29 and the likelihood function. Hence, the JPD reduces to

$$\pi^*(\alpha, \gamma, \eta | \underline{x}) = K\varphi(\alpha, \gamma, \eta) \prod_{i=1}^{n} \frac{\alpha\gamma x_i^{\alpha-1} e^{-\gamma x_i^{\alpha}} \left(e^{(1-e^{-\gamma x_i^{\alpha}})} + \eta\right)}{e - \bar{\eta}}, \tag{31}$$

**Table 5. Simulation results of the eight estimation methods for $\phi = (\alpha = 2.75, \gamma = 0.50, \eta = 1.50)^{\top}$.**

| n | Measures | Est. Par. | MLEs | MPSEs | LSEs | CVMEs | WLSEs | PCEs | ADEs | RADEs |
|---|---|---|---|---|---|---|---|---|---|---|
| 20 | AVEs | $\hat{\alpha}$ | 2.90343 | 2.41702 | 2.49703 | 2.77100 | 2.45786 | 2.41480 | 2.66096 | 2.97477 |
| | | $\hat{\gamma}$ | 0.51303 | 0.52863 | 0.49951 | 0.48743 | 0.50168 | 0.51204 | 0.47782 | 0.57411 |
| | | $\hat{\eta}$ | 1.50963 | 1.40434 | 1.45476 | 1.53200 | 1.46588 | 1.43369 | 1.52228 | 1.42453 |
| | MSEs | $\hat{\alpha}$ | 2.29599 | 2.05640 | 3.30807 | 3.36388 | 2.72663 | 2.20354 | 2.09744 | 3.74722 |
| | | $\hat{\gamma}$ | 0.21174 | 0.20820 | 0.24700 | 0.24592 | 0.23480 | 0.22037 | 0.20703 | 0.24592 |
| | | $\hat{\eta}$ | 0.50532 | 0.48921 | 1.02482 | 1.03259 | 0.78388 | 0.57168 | 0.52528 | 0.81807 |
| | ABs | $\hat{\alpha}$ | 1.51525 | 1.43402 | 1.81881 | 1.83409 | 1.65125 | 1.48443 | 1.44826 | 1.93577 |
| | | $\hat{\gamma}$ | 0.46016 | 0.45629 | 0.49699 | 0.49591 | 0.48456 | 0.46944 | 0.45500 | 0.49591 |
| | | $\hat{\eta}$ | 0.71086 | 0.69943 | 1.01233 | 1.01616 | 0.88537 | 0.75610 | 0.72476 | 0.90447 |
| | MREs | $\hat{\alpha}$ | 0.55100 | 0.52146 | 0.66139 | 0.66694 | 0.60045 | 0.53979 | 0.52664 | 0.70392 |
| | | $\hat{\gamma}$ | 0.92031 | 0.91259 | 0.99397 | 0.99181 | 0.96912 | 0.93887 | 0.91001 | 0.99181 |
| | | $\hat{\eta}$ | 0.47390 | 0.46629 | 0.67489 | 0.67744 | 0.59025 | 0.50407 | 0.48318 | 0.60298 |
| 50 | AVEs | $\hat{\alpha}$ | 2.77955 | 2.56534 | 2.64386 | 2.75591 | 2.65465 | 2.57022 | 2.73965 | 2.83955 |
| | | $\hat{\gamma}$ | 0.49572 | 0.51715 | 0.49581 | 0.49103 | 0.50758 | 0.50566 | 0.49928 | 0.53963 |
| | | $\hat{\eta}$ | 1.52367 | 1.45162 | 1.49119 | 1.51101 | 1.47626 | 1.47207 | 1.50397 | 1.45502 |
| | MSEs | $\hat{\alpha}$ | 0.81967 | 0.72572 | 1.56623 | 1.58154 | 1.06035 | 0.86569 | 0.98967 | 1.64580 |
| | | $\hat{\gamma}$ | 0.10223 | 0.09701 | 0.17341 | 0.17343 | 0.13487 | 0.10754 | 0.11945 | 0.16034 |
| | | $\hat{\eta}$ | 0.15631 | 0.14482 | 0.35480 | 0.36260 | 0.22922 | 0.17378 | 0.19277 | 0.28666 |
| | ABs | $\hat{\alpha}$ | 0.90536 | 0.85189 | 1.25149 | 1.25759 | 1.02973 | 0.93043 | 0.99482 | 1.28289 |
| | | $\hat{\gamma}$ | 0.31974 | 0.31147 | 0.41643 | 0.41645 | 0.36725 | 0.32793 | 0.34562 | 0.40043 |
| | | $\hat{\eta}$ | 0.39536 | 0.38056 | 0.59565 | 0.60216 | 0.47876 | 0.41688 | 0.43906 | 0.53540 |
| | MREs | $\hat{\alpha}$ | 0.32922 | 0.30978 | 0.45509 | 0.45731 | 0.37445 | 0.33834 | 0.36175 | 0.46650 |
| | | $\hat{\gamma}$ | 0.63948 | 0.62294 | 0.83285 | 0.83290 | 0.73450 | 0.65586 | 0.69124 | 0.80086 |
| | | $\hat{\eta}$ | 0.26357 | 0.25370 | 0.39710 | 0.40144 | 0.31918 | 0.27792 | 0.29271 | 0.35694 |
| 100 | AVEs | $\hat{\alpha}$ | 2.75884 | 2.65096 | 2.67985 | 2.74655 | 2.70362 | 2.61822 | 2.75121 | 2.83738 |
| | | $\hat{\gamma}$ | 0.48862 | 0.51175 | 0.50879 | 0.50175 | 0.49765 | 0.50288 | 0.49732 | 0.53902 |
| | | $\hat{\eta}$ | 1.51225 | 1.46516 | 1.48695 | 1.50277 | 1.49492 | 1.48220 | 1.50561 | 1.46262 |
| | MSEs | $\hat{\alpha}$ | 0.40105 | 0.37296 | 0.80667 | 0.89662 | 0.52365 | 0.41959 | 0.49604 | 0.84350 |
| | | $\hat{\gamma}$ | 0.05326 | 0.04885 | 0.10198 | 0.11409 | 0.07314 | 0.05913 | 0.06699 | 0.09360 |
| | | $\hat{\eta}$ | 0.07629 | 0.06701 | 0.15929 | 0.18465 | 0.10658 | 0.08248 | 0.09744 | 0.12932 |
| | ABs | $\hat{\alpha}$ | 0.63329 | 0.61071 | 0.89815 | 0.94690 | 0.72364 | 0.64776 | 0.70430 | 0.91842 |
| | | $\hat{\gamma}$ | 0.23077 | 0.22101 | 0.31935 | 0.33777 | 0.27045 | 0.24317 | 0.25882 | 0.30594 |
| | | $\hat{\eta}$ | 0.27621 | 0.25887 | 0.39911 | 0.42971 | 0.32647 | 0.28719 | 0.31216 | 0.35961 |
| | MREs | $\hat{\alpha}$ | 0.23029 | 0.22208 | 0.32660 | 0.34433 | 0.26314 | 0.23555 | 0.25611 | 0.33397 |
| | | $\hat{\gamma}$ | 0.46154 | 0.44202 | 0.63869 | 0.67554 | 0.54091 | 0.48634 | 0.51764 | 0.61187 |
| | | $\hat{\eta}$ | 0.18414 | 0.17258 | 0.26607 | 0.28647 | 0.21765 | 0.19146 | 0.20811 | 0.23974 |
| 200 | AVEs | $\hat{\alpha}$ | 2.77445 | 2.71123 | 2.72260 | 2.76345 | 2.69839 | 2.69544 | 2.74485 | 2.76819 |
| | | $\hat{\gamma}$ | 0.50091 | 0.50907 | 0.49843 | 0.50361 | 0.49287 | 0.49549 | 0.50214 | 0.50091 |
| | | $\hat{\eta}$ | 1.50570 | 1.48055 | 1.49749 | 1.49556 | 1.50717 | 1.49709 | 1.49735 | 1.49493 |
| | MSEs | $\hat{\alpha}$ | 0.18004 | 0.18451 | 0.43792 | 0.42650 | 0.26449 | 0.19660 | 0.24555 | 0.46385 |
| | | $\hat{\gamma}$ | 0.02591 | 0.02482 | 0.05983 | 0.05892 | 0.03678 | 0.02979 | 0.03342 | 0.05272 |
| | | $\hat{\eta}$ | 0.03415 | 0.03308 | 0.08616 | 0.07948 | 0.04875 | 0.03874 | 0.04337 | 0.06642 |
| | ABs | $\hat{\alpha}$ | 0.42431 | 0.42954 | 0.66176 | 0.65307 | 0.51429 | 0.44340 | 0.49553 | 0.68107 |
| | | $\hat{\gamma}$ | 0.16095 | 0.15754 | 0.24460 | 0.24273 | 0.19177 | 0.17259 | 0.18281 | 0.22961 |
| | | $\hat{\eta}$ | 0.18479 | 0.18188 | 0.29353 | 0.28192 | 0.22079 | 0.19683 | 0.20826 | 0.25772 |
| | MREs | $\hat{\alpha}$ | 0.15430 | 0.15620 | 0.24064 | 0.23748 | 0.18701 | 0.16124 | 0.18019 | 0.24766 |
| | | $\hat{\gamma}$ | 0.32191 | 0.31508 | 0.48920 | 0.48545 | 0.38354 | 0.34518 | 0.36562 | 0.45923 |
| | | $\hat{\eta}$ | 0.12319 | 0.12125 | 0.19568 | 0.18795 | 0.14719 | 0.13122 | 0.13884 | 0.17181 |

*(Continued)*

**Table 5.** (*Continued*)

| n | Measures | Est. Par. | MLEs | MPSEs | LSEs | CVMEs | WLSEs | PCEs | ADEs | RADEs |
|---|----------|-----------|------|-------|------|-------|-------|------|------|-------|
| 400 | AVEs | $\hat{\alpha}$ | 2.74461 | 2.73444 | 2.73143 | 2.76710 | 2.73520 | 2.72389 | 2.74380 | 2.76926 |
| | | $\hat{\gamma}$ | 0.49791 | 0.50565 | 0.50117 | 0.50261 | 0.49341 | 0.50161 | 0.50207 | 0.50469 |
| | | $\hat{\eta}$ | 1.50408 | 1.48832 | 1.49534 | 1.49765 | 1.50456 | 1.49305 | 1.49595 | 1.49604 |
| | MSEs | $\hat{\alpha}$ | 0.08983 | 0.08447 | 0.21277 | 0.22661 | 0.13132 | 0.10080 | 0.13266 | 0.22663 |
| | | $\hat{\gamma}$ | 0.01305 | 0.00995 | 0.02933 | 0.03072 | 0.01866 | 0.01369 | 0.01859 | 0.02594 |
| | | $\hat{\eta}$ | 0.01691 | 0.01378 | 0.03931 | 0.03941 | 0.02423 | 0.01750 | 0.02412 | 0.03276 |
| | ABs | $\hat{\alpha}$ | 0.29971 | 0.29063 | 0.46127 | 0.47604 | 0.36238 | 0.31748 | 0.36423 | 0.47606 |
| | | $\hat{\gamma}$ | 0.11423 | 0.09976 | 0.17125 | 0.17528 | 0.13660 | 0.11700 | 0.13634 | 0.16105 |
| | | $\hat{\eta}$ | 0.13005 | 0.11741 | 0.19828 | 0.19852 | 0.15564 | 0.13228 | 0.15530 | 0.18100 |
| | MREs | $\hat{\alpha}$ | 0.10899 | 0.10568 | 0.16774 | 0.17310 | 0.13178 | 0.11545 | 0.13245 | 0.17311 |
| | | $\hat{\gamma}$ | 0.22845 | 0.19951 | 0.34249 | 0.35056 | 0.27319 | 0.23399 | 0.27268 | 0.32210 |
| | | $\hat{\eta}$ | 0.08670 | 0.07827 | 0.13218 | 0.13235 | 0.10376 | 0.08819 | 0.10353 | 0.12067 |

where *K* is defined by

$$K^{-1} = \int_0^\infty \int_0^\infty \int_0^\infty \varphi(\alpha, \gamma, \eta) \prod_{i=1}^n \frac{\alpha \gamma x_i^{\alpha-1} e^{-\gamma x_i^\alpha} \left( e^{(1-e^{-\gamma x_i^\alpha})} + \eta \right)}{e - \bar{\eta}} \, d\alpha \, d\gamma \, d\eta. \tag{32}$$

Eq (31) shows that there is no closed form for the Bayes estimates (BEs) under the LFs in Table 7, hence we will use the *MCMC* procedure based on 10, 000 replicates to obtain the BEs. We calculate the BEs of the TIEx-W parameters under different LFs which are mentioned in Table 7. The Bayesian point and interval estimation and posterior risk for the insurance data are listed in Table 8. Table 9 lists 95% credible and *HPD* intervals for the TIEx-W parameters. The posterior samples are extracted using Gibbs sampling technique. The MCMC iterations of $\alpha$, $\gamma$ and $\eta$ are, respectively, plotted. These summary plots are provided in Figs 5–7. In summary, the BEs of the TIEx-W parameters are consistent especially under the SELF in terms of their lowest risks.

Table 10 displays the parameter estimates under various estimation methods with Bayesian estimation and the goodness-of-fit statistics for insurance data. Furthermore, the histogram of the fitted TIEx-W model under various estimation methods with Bayesian estimation for insurance data are displayed in Fig 8. Furthermore, Fig 8 shows the fitted densities and

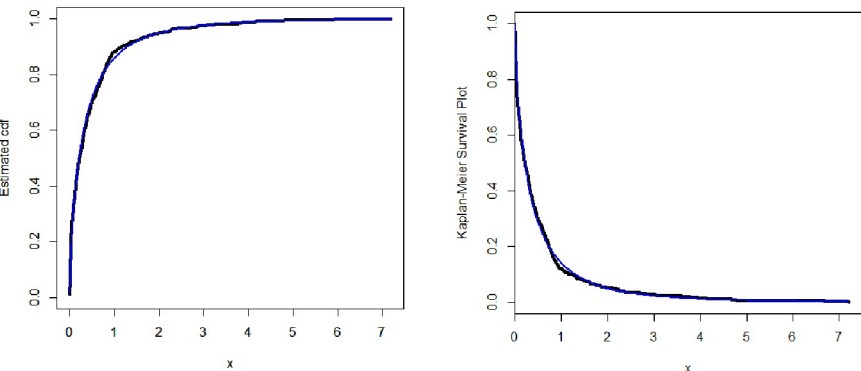

**Fig 3. Fitted cdf and KMS plots of the TIEx-W distribution for insurance data.**

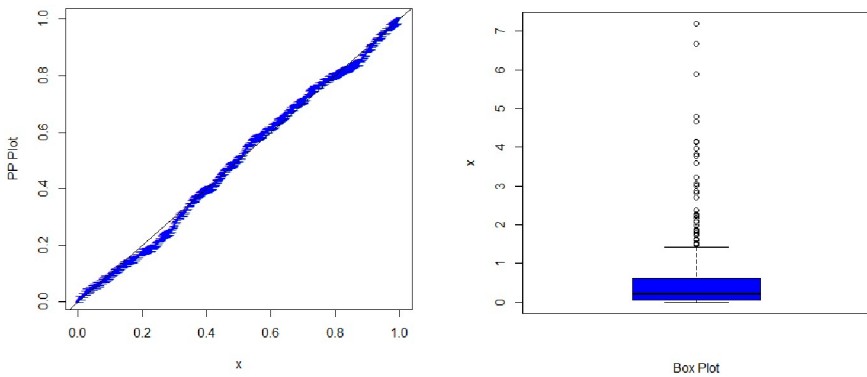

**Fig 4. PP plot of the TIEx-W distribution and box plot for insurance data.**

**Table 6. MLEs (SEs in parentheses), K-S statistics and *p*-values of the TIEx-W and other competing distributions, for the insurance data set.**

| Model | Estimates | | | | | Goodness-of-fit | |
|---|---|---|---|---|---|---|---|
| | $\hat{\alpha}$ | $\hat{\gamma}$ | $\hat{\eta}$ | $\hat{a}$ | $\hat{b}$ | K-S (Stat) | K-S (*p*-value) |
| **TIExW** | 0.67414 | 1.91856 | 12007.63364 | | | **0.03590** | **0.50674** |
| | (0.02246) | (0.08470) | (16777.22997) | | | | |
| Wi | 0.67415 | 0.38043 | | | | 0.03590 | 0.50673 |
| | (0.02246) | (0.02598) | | | | | |
| LiW | 0.00690 | 0.67418 | 55.94897 | | | 0.03593 | 0.50572 |
| | (0.00196) | (0.02251) | (12.51911) | | | | |
| EW | 1.82994 | 0.15255 | | 0.49259 | | 0.03903 | 0.39948 |
| | (0.45802) | (0.06667) | | (0.06312) | | | |
| KwW | 15.73377 | 0.11465 | | 11.20819 | 51.02618 | 0.03938 | 0.38833 |
| | (77.94853) | (0.06407) | | (9.77843) | (69.77317) | | |
| ALW | 0.08038 | 0.89733 | 1.05798 | | | 0.04081 | 0.34504 |
| | (0.08479) | (0.11226) | (0.27515) | | | | |
| Bur | 0.81321 | 2.88579 | | | | 0.04536 | 0.22931 |
| | (0.02661) | (0.12984) | | | | | |
| Lo | 0.36558 | 1.56880 | | | | 0.07889 | 0.00287 |
| | (0.07480) | (0.20762) | | | | | |
| BW | 0.16375 | 0.68594 | 85.85773 | 0.87108 | | 0.11615 | 0.41433 |
| | (0.22136) | (0.35314) | (131.94042) | (0.76148) | | | |
| APTW | 1.20911 | 0.01502 | 1652.58873 | | | 0.13768 | 0.22159 |
| | (0.09703) | (0.00654) | (1540.20546) | | | | |
| AmpaduAPTW | 1.22798 | 0.01352 | 1045.39113 | | | 0.14173 | 0.19440 |
| | (0.09547) | (0.00580) | (921.59604) | | | | |
| ExAPTW | 1.25152 | 0.01184 | 594.89942 | | | 0.14625 | 0.16717 |
| | (0.09267) | (0.00494) | (503.15363) | | | | |

**Table 7. Five LFs and their Bayes estimators and posterior risk.**

| Loss function (LF) | Formula | Bayes estimators | Posterior risk |
|---|---|---|---|
| squared error LF (SELF) | $L_1 = (\theta - d)^2$ | $E(\theta\|x)$ | $Var(\theta\|x)$ |
| weighted SELF (WSELF) | $L_2 = \frac{(\theta-d)^2}{\theta}$ | $(E(\theta^{-1}\|x))^{-1}$ | $E(\theta\|x) - (E(\theta^{-1}\|x))^{-1}$ |
| modified SELF (MSELF) | $L_3 = \left(1 - \frac{d}{\theta}\right)^2$ | $\frac{E(\theta^{-1}\|x)}{E(\theta^{-2}\|x)}$ | $1 - \frac{E(\theta^{-1}\|x)^2}{E(\theta^{-2}\|x)}$ |
| K-LF (KLF) | $L_4 = \left(\sqrt{\frac{d}{\theta}} - \sqrt{\frac{\theta}{d}}\right)$ | $\sqrt{\frac{E(\theta\|x)}{E(\theta^{-1}\|x)}}$ | $2\left(\sqrt{E(\theta\|x)E(\theta^{-1}\|x)} - 1\right)$ |
| precautionary LF (PLF) | $L_5 = \frac{(\theta-d)^2}{d}$ | $\sqrt{E(\theta^2\|x)}$ | $2\left(\sqrt{E(\theta^2\|x)} - E(\theta\|x)\right)$ |

**Table 8. The BEs and posterior risks of the TIEx-W the parameters for insurance data.**

| Data | Insurance data | | | | | |
|---|---|---|---|---|---|---|
| Bayes | $\hat{\alpha}$ | | $\hat{\gamma}$ | | $\hat{\eta}$ | |
| Loss functions | Estimate | Risk | Estimate | Risk | Estimate | Risk |
| SELF | 0.41240 | 5.08798e-06 | 1.93621 | 0.00105 | 0.02237 | 0.00195 |
| WSELF | 0.41230 | 1.23323e-05 | 1.93567 | 0.00054 | 0.00853 | 0.01384 |
| MSELF | 0.41210 | 2.98955e-05 | 1.93513 | 0.00028 | 0.00261 | 0.69415 |
| PLF | 0.41220 | 1.23376e-05 | 1.93648 | 0.00054 | 0.04950 | 0.05426 |
| KLF | 0.41240 | 2.99084e-05 | 1.93594 | 0.00028 | 0.01381 | 1.23843 |

distribution functions of the TIEx-W under various estimation methods with Bayesian estimation for insurance data.

## 9 Concluding remarks

In the present paper, we have introduced a new class of heavy-tailed distributions allowing closed form expressions for distribution function and some of its basic properties. The proposed class is called type-I extended-F (TIEx-F) family and one of its special sub-models, called the TIEx-Weibull (TIEx-W) distribution, is addressed. The TIEx-W parameters are obtained based on eight methods of estimation, and a detailed simulation study is provided. Based on our study, we conclude that the maximum likelihood method outperforms all other classical estimation methods. Hence, it is recommended to estimate the parameters of the TIEx-Weibull distribution. The applicability of the TIEx-W distribution has been illustrated using an insurance data set. The insurance data is fitted using the TIEx-W distribution and other competing models. The results illustrate that the TIEx-W distribution provides better fit as compared to competing distributions. The Bayesian analysis based on real-life insurance data is also discussed under five loss functions. The analysis shows that the Bayesian estimates of the

**Table 9. Credible and *HPD* intervals of the TIEx-W parameters $\alpha$, $\gamma$ and $\eta$ for insurance data.**

| Parameter | Credible interval | HPD interval |
|---|---|---|
| $\alpha$ | (0.4108, 0.4139) | (0.4077, 0.4163) |
| $\gamma$ | (1.867, 1.996) | (0.0085, 0.0299) |
| $\eta$ | (0.008, 0.024) | (0.0004, 0.0451) |

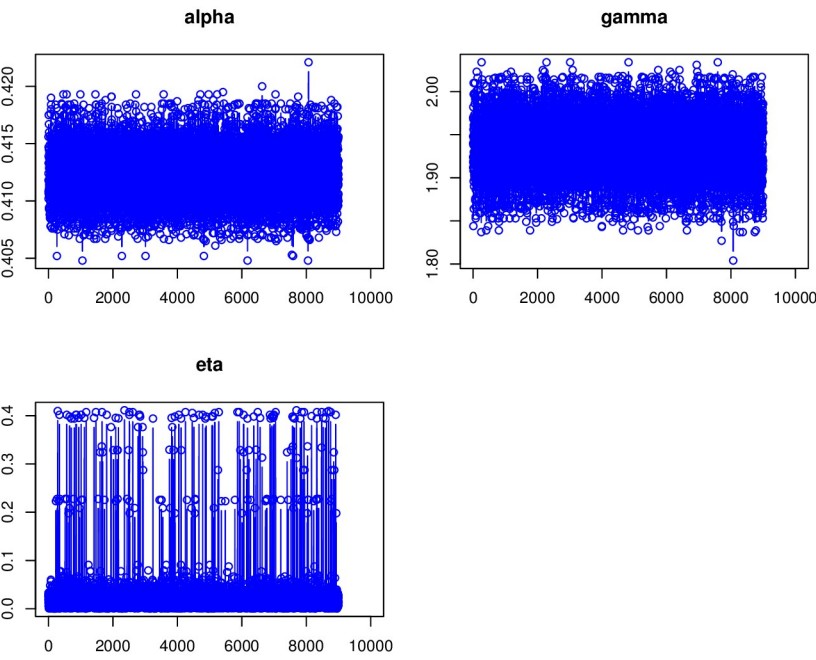

**Fig 5. Trace plots of each parameter of TIEx-W distribution.**

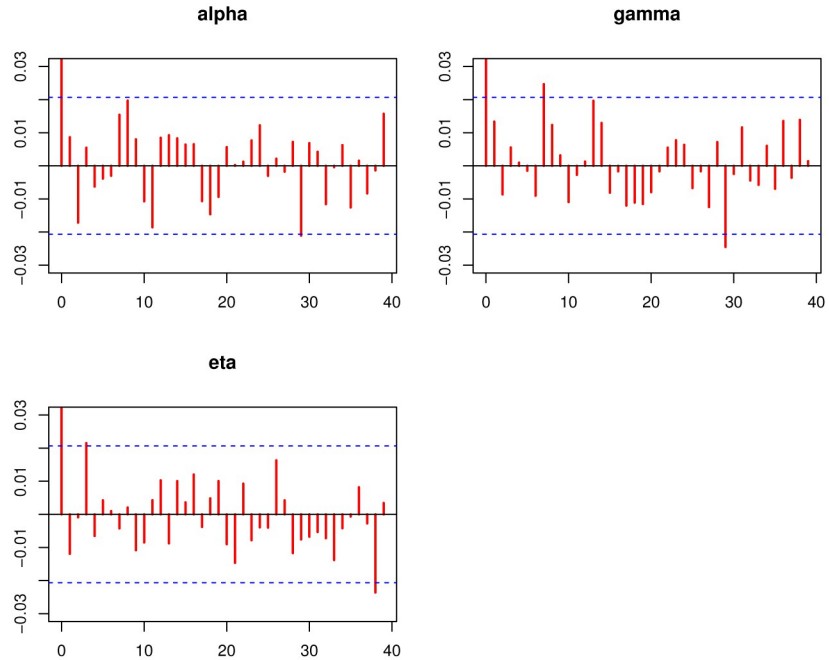

**Fig 6. Autocorrelation plots of each parameter of TIEx-W distribution.**

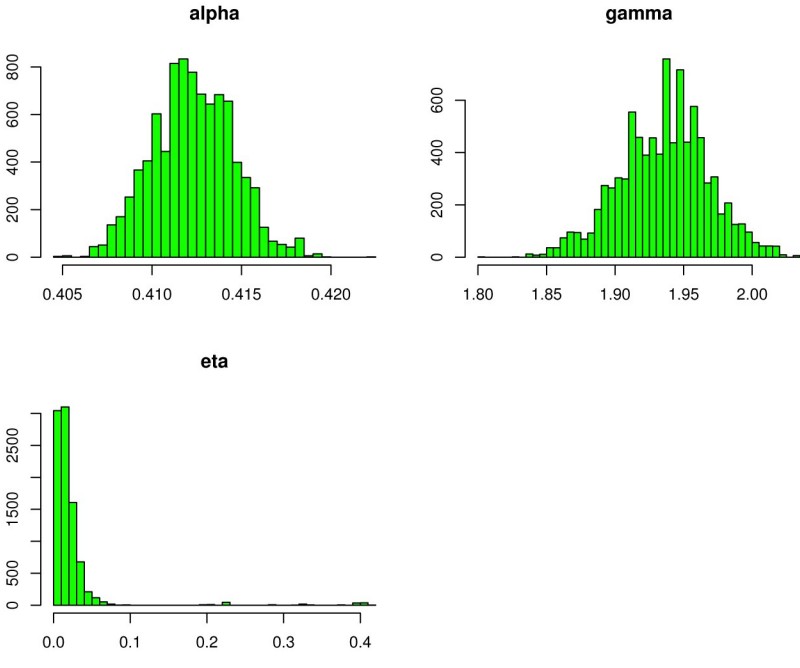

**Fig 7. Histogram plots of each parameter of TIEx-W distribution.**

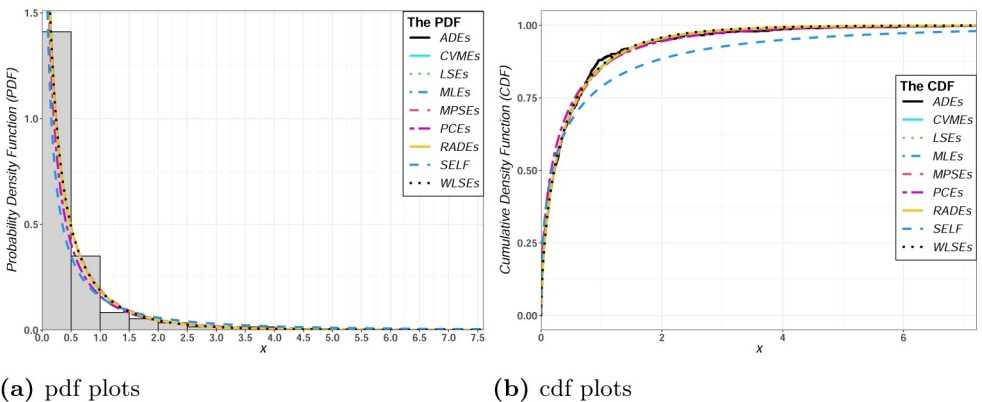

(a) pdf plots

(b) cdf plots

**Fig 8. Fitted densities and distribution functions of the TIEx-W under various estimation methods with Bayesian estimation for insurance data.**

TIEx-W parameters are consistent especially under the squared error loss function in terms of their lowest risks.

## Acknowledgments

The authors would like to thank the Editorial Board, and three reviewers for their constructive comments and suggestions which greatly improved the final version of this manuscript.

**Table 10. The parameter estimates under various estimation methods with Bayesian estimation, including the K-S statistic and K-S p-value for insurance data.**

| Method | $\hat{\alpha}$ | $\hat{\gamma}$ | $\hat{\eta}$ | K-S (Stat) | K-S (p-value) |
|---|---|---|---|---|---|
| ADEs | 0.67373 | 1.95095 | 83412.18951 | 0.03194 | 0.65662 |
| CVMEs | 0.65421 | 2.06502 | 5.20179 | 0.03106 | 0.69048 |
| LSEs | 0.65238 | 2.05969 | 5.20169 | 0.03180 | 0.66193 |
| MLEs | 0.67415 | 1.91855 | 12007.63364 | 0.03590 | 0.50663 |
| MPSEs | 0.65197 | 2.01517 | 5.20248 | 0.03837 | 0.42086 |
| PCEs | 0.55892 | 2.05527 | 5.20106 | 0.07296 | 0.00739 |
| RADEs | 0.64834 | 2.05464 | 5.20160 | 0.03247 | 0.63619 |
| SELF | 0.41240 | 1.93621 | 0.02237 | 0.10145 | 0.00004 |
| WLSEs | 0.68402 | 1.97246 | 83551.79860 | 0.03584 | 0.50900 |

## Author Contributions

**Conceptualization:** Nada M. Alfaer, Omid Kharazmi, Zubair Ahmad, Ahmed Z. Afify.

**Formal analysis:** Sarah A. Bandar, Zubair Ahmad.

**Investigation:** Sarah A. Bandar, Omid Kharazmi, Hazem Al-Mofleh, Ahmed Z. Afify.

**Methodology:** Nada M. Alfaer, Sarah A. Bandar, Omid Kharazmi, Hazem Al-Mofleh, Zubair Ahmad.

**Project administration:** Ahmed Z. Afify.

**Resources:** Nada M. Alfaer, Omid Kharazmi.

**Software:** Omid Kharazmi, Hazem Al-Mofleh, Zubair Ahmad.

**Supervision:** Ahmed Z. Afify.

**Validation:** Sarah A. Bandar.

**Visualization:** Nada M. Alfaer, Sarah A. Bandar.

**Writing – original draft:** Nada M. Alfaer, Omid Kharazmi, Zubair Ahmad, Ahmed Z. Afify.

**Writing – review & editing:** Nada M. Alfaer, Sarah A. Bandar, Hazem Al-Mofleh, Ahmed Z. Afify.

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
