## [Decision Letter · Decision Letter 0]

23 Aug 2022

PONE-D-22-21984Classical and Bayesian Estimation for Type-I Extended-F Family with an Actuarial ApplicationPLOS ONE

Dear Dr. Afify,

Thank you for submitting your manuscript to PLOS ONE. After careful consideration, we feel that it has merit but does not fully meet PLOS ONE’s publication criteria as it currently stands. Therefore, we invite you to submit a revised version of the manuscript that addresses the points raised during the review process.

We look forward to receiving your revised manuscript.

Kind regards,

Srinivasa Rao Gadde, Ph.D.

Academic Editor

PLOS ONE

Journal Requirements:

3. Please ensure that you refer to Figures 4 to 6 in your text as, if accepted, production will need this reference to link the reader to the figure.

4. We note you have included a table to which you do not refer in the text of your manuscript. Please ensure that you refer to Tables 1, 2, 6, 8, 9, and 10 in your text; if accepted, production will need this reference to link the reader to the Table.

Additional Editor Comments:

Dear Author,

Please revise and submit the manuscript. Remember that you have to respond to all comments given the reviewers. Please prepare response matrix for each reviewers and revised manuscript within 10 days.

with best wishes

Reviewers' comments:

Reviewer's Responses to Questions

**Comments to the Author**

1. Is the manuscript technically sound, and do the data support the conclusions?

Reviewer #1: Yes

Reviewer #2: Partly

Reviewer #3: Yes

2. Has the statistical analysis been performed appropriately and rigorously? 

Reviewer #1: Yes

Reviewer #2: Yes

Reviewer #3: Yes

3. Have the authors made all data underlying the findings in their manuscript fully available?

Reviewer #1: Yes

Reviewer #2: Yes

Reviewer #3: Yes

4. Is the manuscript presented in an intelligible fashion and written in standard English?

Reviewer #1: No

Reviewer #2: Yes

Reviewer #3: Yes

5. Review Comments to the Author

Reviewer #1: 1.The introduction must be separate from the methodology

2.The conclusion must be well written and organized

3.Some tables do not cite; they must be cited (“The parameters estimate …. Table??. The goodness… in Table??.”)

4.An acronym must be added.

5.The writing style of the manuscript have been not follow the guidelines

6.The discussion part is essential. But not written in the manuscript

7.Resnick, S.I. (1997). Discussion of the Danish data on large fire insurance losses. 268 ASTIN Bulletin: The Journal of the IAA, 27, 139-151.not urgent

8.McNeil, A. J. (1997). Estimating the tails of loss severity distributions using 261 extreme value theory. ASTIN Bulletin: The Journal of the IAA, 27(1), 117-137: not urgent

Reviewer #2: This study proposes a new family of distributions named Type-I Extended-F Family (TIEx-F) with a specific focus on modelling insurance data. Some properties and numerous classical and Bayesian estimation approaches are introduced for estimating the parameters of a special case, the TIEx-Weibull (TIEx-W) model. Finally, the author demonstrate the potentials of the model by applying it to a single insurance datasets. The authors have performed a good job, however, I have some issues and suggestions that may enhance the current version of the manuscript. See the attached file for the comments

Reviewer #3: The authors have put in considerable effort in explaining their research aim and finding but need to add some comments on the tables and charts rather than leaving them blank. The functional for of the estimators should also be provided even if he procedure can be verified by readers.

6. PLOS authors have the option to publish the peer review history of their article (what does this mean?). If published, this will include your full peer review and any attached files.

Reviewer #1: No

Reviewer #2: No

Reviewer #3: No

---

## [Author Response · Author response to Decision Letter 0]

7 Sep 2022

Response Letter

Respected Editor

Enclosed herewith are the pdf and latex files of the revised version of our paper entitled "Classical and Bayesian Estimation for Type-I Extended-F Family with an Actuarial Application ", which we hope you will find now satisfactory. 

First, we would like to thank the Editor and three reviewers for his very constructive comments. In the revised version all suggestions and comments have been taken into account and addressed. All corrections and modifications are incorporated in the revised version and highlighted in magenta color. 

We now answer the comments made by the Editor and the three reviewers in the order they appeared in the reports.

Journal Requirements: 

Answer: We have prepared the paper according to PLOS ONE's style requirements.

Answer: Thanks. We corrected them.

3. Please ensure that you refer to Figures 4 to 6 in your text as, if accepted, production will need this reference to link the reader to the figure.

Answer: We double checked and cited all figures and tables in the text.

4. We note you have included a table to which you do not refer in the text of your manuscript. Please ensure that you refer to Tables 1, 2, 6, 8, 9, and 10 in your text; if accepted, production will need this reference to link the reader to the Table.

Answer: We double checked and cited all figures and tables in the text.

Answer: Thanks for this comment. We have removed two references as suggested by Reviewer #1. Some references has been added in the application section (Section 7) because we have added some new models in the application to compare them with the TIEx-W model as suggested by Reviewer #2. All changes in the revised version are highlighted in magenta color.

Reviewers' comments:

Reviewer #1: 

1. The introduction must be separate from the methodology.

Answer: Thanks for this suggestion. We have improved accordingly. We have added a new section entitled ‘’The TIEx-F Family’’.

2. The conclusion must be well written and organized.

Answer: Thanks for this suggestion. The conclusion section is corrected accordingly.

3. Some tables do not cite; they must be cited (“The parameters estimate …. Table??. The goodness… in Table??.”)

Answer: Thanks for careful reading. All tables and figures are now cited in the text.

4. An acronym must be added.

Answer: Thanks for this comment. We have added a list of abbreviations at the end of the paper.

5. The writing style of the manuscript have been not follow the guidelines.

Answer: We have double checked and improved the writing style throughout the paper.

6. The discussion part is essential. But not written in the manuscript.

Answer: Thanks for this comment. We have improved the conclusion section to include the discussion as you suggested in your Comment #2.

7. Resnick, S.I. (1997). Discussion of the Danish data on large fire insurance losses. 268 ASTIN Bulletin: The Journal of the IAA, 27, 139-151. not urgent

8. McNeil, A. J. (1997). Estimating the tails of loss severity distributions using 261 extreme value theory. ASTIN Bulletin: The Journal of the IAA, 27(1), 117-137: not urgent.

Answer: Thanks for this suggestion. We have removed the two references.

Reviewer #2: 

This study proposes a new family of distributions named Type-I Extended-F Family (TIEx-F) with a specific focus on modelling insurance data. Some properties and numerous classical and Bayesian estimation approaches are introduced for estimating the parameters of a special case, the TIEx-Weibull (TIEx-W) model. Finally, the authors demonstrate the potentials of the model by applying it to a single insurance dataset. The authors have performed a good job; however, I have some issues and suggestions that may enhance the current version of the manuscript. 

 There should be more clear motivation of the study.

Answer: Many thanks for this comment. We have improved the motivation accordingly. Please see the introduction section.

 The authors refer the TIExF family "the most tractable for simple analytical expressions" with TIEx-W as a special case, I expect the authors to assess the TIEx-W with other competing non-standard distributions, such as the special cases derived from families defined in Equations (1)-(3). Going by this will explore more the potentials of the proposed family.

Answer: Thanks for this comment. We add the following special case derived from families defined in Equations (1)-(3): APTW, AmpaduAPTW and ExAPTW. We have also added some competing models. Please see Table 6.

 Check the correctness of the sentence and citations in page 3, lines 55-56.

Answer: This sentence is checked and corrected.

 The authors should provide the other basic functions of the TIEx-W distribution, including the survival and hazard rate functions, along with plot(s) showing some possible shapes of the hazard rate function for different settings of the model's parameters. Also present some discussions on the plots and whether the hazard rate function is relevant in modelling insurance data.

Answer: Thanks for this suggestion. We have provided the required discussions, functions, figures, and discussions. Please see Equations (8) and (9), and Figure 2.

 In section 3, more properties of the distribution should be present not only the quantile function and moments.

Answer: Thanks for this comment. We have provided some more properties such as the quantile function, median, shapes of TIEx-W pdf and the order statistics, please see sub-sections 4.2, 4.4 and 4.5.

 Do your quantile function has a closed form expression? If yes, please, write it clearly, and if no, provide a procedure on how to draw random sample from the quantile function.

Answer: Thanks for your comment. Yes, the quantile function has a closed form expression, we provided it in Equation (10) for the TIEx-F family and in Equation (11) for the TIEx-W model.

 Table numbers are missing in the text. Check and correct it.

Answer: Thanks for careful reading. We have checked and corrected them.

 Check and correct the third term of Eq. (15). Not correctly differentiated.

Answer: Thanks for your comment. We have double checked Equation (15). It is correct. We have just moved this term ‘’e^(-γx_k^α )’’ from the numerator ‘’e^(-γx_k^α )’’ to the denominator to be e^(γx_k^α ).

 In section 5, I suggest the authors to include simulation results for the Bayesian approach under the selected loss functions to enable the comparison of all the proposed estimators for the TIEx-W model.

Answer: Thanks for this comment. It is well-known that the Bayesian estimators are better than classical estimators follow using classical methods of estimation. Hence, your suggestion will be a good point for a future paper. In this future work, we may propose the Bayes estimators of the parameters of the TIEx-W distribution for complete and censored samples under different

 loss functions. We can also use different approximation techniques to obtain the Bayes estimates of the parameters.

 In section 7, discuss the reasons/justifications for choosing gamma priors for the three parameters of TIEx-W model.

Answer: Thanks for this comment. It is well-known that the gamma priors provide flexible approach to handle estimation procedure in both informative and non-informative.

 The authors need to provide some discussions on the Bayesian results.

Answer: Thanks for this comment. We have provided the required discussion.

 I also expect to see some comparisons between the results from the classical estimation methods and the Bayesian results either by plotting the density curves, CDF curves for the trained TIEx-W model from all the estimation methods employed or the p-p plots. This is essential to identify the best estimation method(s) in solving real-world problems.

Answer: Thanks for this comment. We have provided a new table which includes the estimates of the TIEx-W parameters using all estimation methods. We have also provided the density plots and PP plots for all methods.

 How the estimated values of the parameters can be translated in the real-life situations?

Answer: Thanks for this comment. Our aim is to show the flexibility of the proposed TIEx-W distribution in modelling insurance data. Based on the fitted TIEx-W model, we can answer some questions such as the probability of having more than X claims per day.

Reviewer #3: 

The authors have put in considerable effort in explaining their research aim and finding but need to add some comments on the tables and charts rather than leaving them blank. The functional for of the estimators should also be provided even if the procedure can be verified by readers.

Answer: Thanks for this comment. We have provided the required comments about tables and figures. The functions for all proposed estimators are provided in the paper. We have also added some new results as suggested by the two other reviewers.

---

## [Editor Report · Decision Letter 1]

19 Sep 2022

Classical and Bayesian Estimation for Type-I Extended-F Family with an Actuarial Application

PONE-D-22-21984R1

Dear Dr. Afify,

We’re pleased to inform you that your manuscript has been judged scientifically suitable for publication and will be formally accepted for publication once it meets all outstanding technical requirements.

Kind regards,

Srinivasa Rao Gadde, Ph.D.

Academic Editor

PLOS ONE

Additional Editor Comments (optional):

The author has responded all my comments as well as reviewer's comments. No new comments.
---

## [Editor Report · Acceptance letter]

29 Sep 2022

PONE-D-22-21984R1 

Classical and Bayesian Estimation for Type-I Extended-F Family with an Actuarial Application 

Dear Dr. Afify:

I'm pleased to inform you that your manuscript has been deemed suitable for publication in PLOS ONE. Congratulations! Your manuscript is now with our production department. 

Kind regards, 

on behalf of

Professor Srinivasa Rao Gadde 

Academic Editor

PLOS ONE